# Power-law population heterogeneity governs epidemic waves

**Jonas Neipel, Jonathan Bauermann, Stefano Bo, Tyler Harmon, Frank Jülicher** *

Max Planck Institute for the Physics of Complex Systems, Dresden, Germany

* julicher@pks.mpg.de

## Abstract

We generalize the Susceptible-Infected-Removed (SIR) model for epidemics to take into account generic effects of heterogeneity in the degree of susceptibility to infection in the population. We introduce a single new parameter corresponding to a power-law exponent of the susceptibility distribution at small susceptibilities. We find that for this class of distributions the gamma distribution is the attractor of the dynamics. This allows us to identify generic effects of population heterogeneity in a model as simple as the original SIR model which is contained as a limiting case. Because of this simplicity, numerical solutions can be generated easily and key properties of the epidemic wave can still be obtained exactly. In particular, we present exact expressions for the herd immunity level, the final size of the epidemic, as well as for the shape of the wave and for observables that can be quantified during an epidemic. In strongly heterogeneous populations, the herd immunity level can be much lower than in models with homogeneous populations as commonly used for example to discuss effects of mitigation. Using our model to analyze data for the SARS-CoV-2 epidemic in Germany shows that the reported time course is consistent with several scenarios characterized by different levels of immunity. These scenarios differ in population heterogeneity and in the time course of the infection rate, for example due to mitigation efforts or seasonality. Our analysis reveals that quantifying the effects of mitigation requires knowledge on the degree of heterogeneity in the population. Our work shows that key effects of population heterogeneity can be captured without increasing the complexity of the model. We show that information about population heterogeneity will be key to understand how far an epidemic has progressed and what can be expected for its future course.

**Data Availability Statement:** We use in our work publicly available data (Referenced as [26] and [34]). The infection data for Germany [26] can be found at https://npgeo-corona-npgeo-de.hub.arcgis.com/, following "data" and "RKI COVID19".

## 1 Introduction

Diseases that spread by transmission between individuals can give rise to epidemic waves that pass through a population [1, 2]. One infected person can infect several others who are susceptible to the infection, characterized by the basic reproduction number $R_0$, initially typically generating an exponential growth of the number of infections. The number of infections reaches a peak and later dies down when there is a sufficient number of individuals that have gained immunity after they recovered from the infection so that further growth is hampered.

The mobility data [34] can be found at https://www.google.com/covid19/mobility. We also provide the datafiles used in our work at https://github.com/jbr4l/SARS-CoV2-PKS.

**Funding:** This work was funded by the Max Planck Society. The funders had no role in study design, data collection and analysis, decision to publish, or preparation of the manuscript. The authors received no specific funding for this work.

**Competing interests:** The authors have declared that no competing interests exist.

The fraction of immune individuals reached at the point when the epidemic starts to recede is called herd immunity [1, 3, 4].

There are big uncertainties as to when and why an epidemic reaches its peak and the levels of herd immunity required [5]. Simple models of infections dynamics predict that for an initially fast growing epidemic most of the population will become infected before the epidemic dies down [1, 3, 6]. It was noted early by William Farr when investigating smallpox and other epidemics that epidemics appear to follow a general time course in the form of a skewed bell shaped curve [7, 8]. They first grow fast, reach a peak and then die down quickly, typically much before the majority of a population has been affected. The fact that an epidemic dies down is usually attributed to the fact that there exists some degree of immunity in the population [9]. The uncertainty about when the peak of an epidemic is reached and why an epidemic dies out even if there remains a large number of still susceptible individuals reveals that the factors that limit an epidemic are not well understood. Furthermore, the effectiveness and impact of mitigation measures such as social distancing to counter a fast growing epidemic are not known.

Simplified models of infection dynamics, such as the classic Susceptible-Infected-Removed (SIR) model have been used for a long time to describe the dynamics of epidemics spreading through a population [1, 3, 6, 10]. Such models capture key features of the epidemic as a non-linear wave with qualitative properties that match observed bell-shaped dynamics of epidemic waves. However, more quantitatively, such models exhibit the robust feature that a quickly growing epidemic does not stop unless the majority of a susceptible population has reached immunity after going through the infection [1]. This raises the question whether important factors are missing in these simple and elegant models. To understand at what conditions and at what levels epidemic waves become self-limiting and die down remains an important challenge. This aspect is also key to understand the role and effectiveness of social distancing measures to influence dynamics of an epidemic wave [10–12].

Simple epidemic models treat the population as effectively consisting of identical individuals. However, individuals in a population can differ widely. The importance of population heterogeneity was put forward to understand smallpox epidemic which could not be captured by simple models [13]. Such heterogeneity has been taken into account by adding details such as introducing several compartments to a model [14] or by introducing distributions of susceptibility [13, 15, 16] or infectiousness [15–17]. It was suggested that population heterogeneity reduces effective herd immunity levels [13, 16, 18, 19].

In this paper, we present a generalization of the SIR model that takes into account effects of population heterogeneity. We show here that effects of heterogeneity can be added without losing the simplicity of the SIR model and keeping its mathematical structure. We introduce a single new parameter, the susceptibility exponent $\alpha$, which characterizes a generic power-law heterogeneity in the distribution of infection susceptibilities of the population. Power laws are often found in nonlinear and complex systems [20–23]. In the present context, power laws could be expected for example based on a variability of immune responses of different individuals which could imply a wide variability in the efficiency of the transmission of an infection [24, 25]. Furthermore, population heterogeneity could be relevant at very different scales, from the immune response of cells to the behaviors of individuals that affect infection rates. Such as broad range of relevant scales could give rise to approximately scale free properties or power laws.

In the heterogeneous SIR model proposed here, the qualitative behaviors of the epidemic wave are unchanged. However, as a function of the parameter $\alpha$, the wave can become self-limited at much lower levels of infected individuals as compared to the classic SIR model. In the limit of large $\alpha$ we recover the classic SIR model of homogeneous populations. For smaller $\alpha$

we find that the number of infections at the peak and the cumulative number of infections after the epidemic has passed can be strongly reduced. Our work has implications for the concept of herd immunity and clarifies that herd immunity cannot be discussed independently of population heterogeneity.

We discuss the dynamics of the SARS-CoV-2 pandemics using the heterogeneous SIR model applied to data on reported infection numbers and COVID-19 associated deaths in Germany [26]. We estimate parameter values including the susceptibility exponent $\alpha$ and show that the time course observed in Germany is compatible with different scenarios ranging from a homogeneous population strongly affected by mitigation to a self-limited epidemic wave in a heterogeneous population where social distancing measures play a minor role.

## II. The Susceptible-Infected-Removed model

The Susceptible-Infected-Removed (SIR) model captures key features of a spreading epidemic as a mean field theory based on pair-wise interactions between infected and susceptible individuals. This model captures generic and robust features without aiming to describe specific details. In the presence of $I$ infected individuals in a population of $N$ individuals, the infection can be transmitted to susceptible individuals. They stay infectious during an average time $\gamma^{-1}$ after which they no longer contribute to infections. The number of susceptible individuals $S$ and the number of infected individuals $I$ obey

$$\dot{S} = -\beta \bar{x} \frac{IS}{N} \tag{1}$$

$$\dot{I} = \beta \bar{x} \frac{IS}{N} - \gamma I \quad , \tag{2}$$

where the dots denote time derivatives. The infection rate is denoted $\beta$ and can in general depend on time $t$. This time dependence could correspond to seasonal changes or mitigation measures [10, 12, 27]. The infection rate can be modulated by the average infection susceptibility $\bar{x}$ which we introduce to capture effects of population heterogeneity discussed below. In the classical SIR model $\bar{x} = 1$. The number of removed (or recovered) individuals is given by $N - S - I$ and the cumulative number of infections is $C = N - S$. A key parameter is the basic reproduction number

$$R_0 = \frac{\beta}{\gamma} \quad , \tag{3}$$

which denotes the average number of new infections generated by an infected individual. The growth rate of infections is $\dot{I}/I = \lambda(t) = \gamma(R(t) - 1)$, where $R(t) = \beta \bar{x} S/(N\gamma)$ is a time dependent reproduction number.

The time course of an epidemic is often provided as the number of new cases per day. This corresponds to the rate of new infections per unit time

$$J = \beta \bar{x} \frac{IS}{N} \tag{4}$$

with $J = \dot{C} = -\dot{S}$ and $R = J/(\gamma I)$.

## A. Infection dynamics in homogeneous populations

In the simple case of a homogeneous population, all individuals have the same degree of susceptibility, $x = 1$ and the population average of $x$ is $\bar{x} = 1$ independent of time. This is the

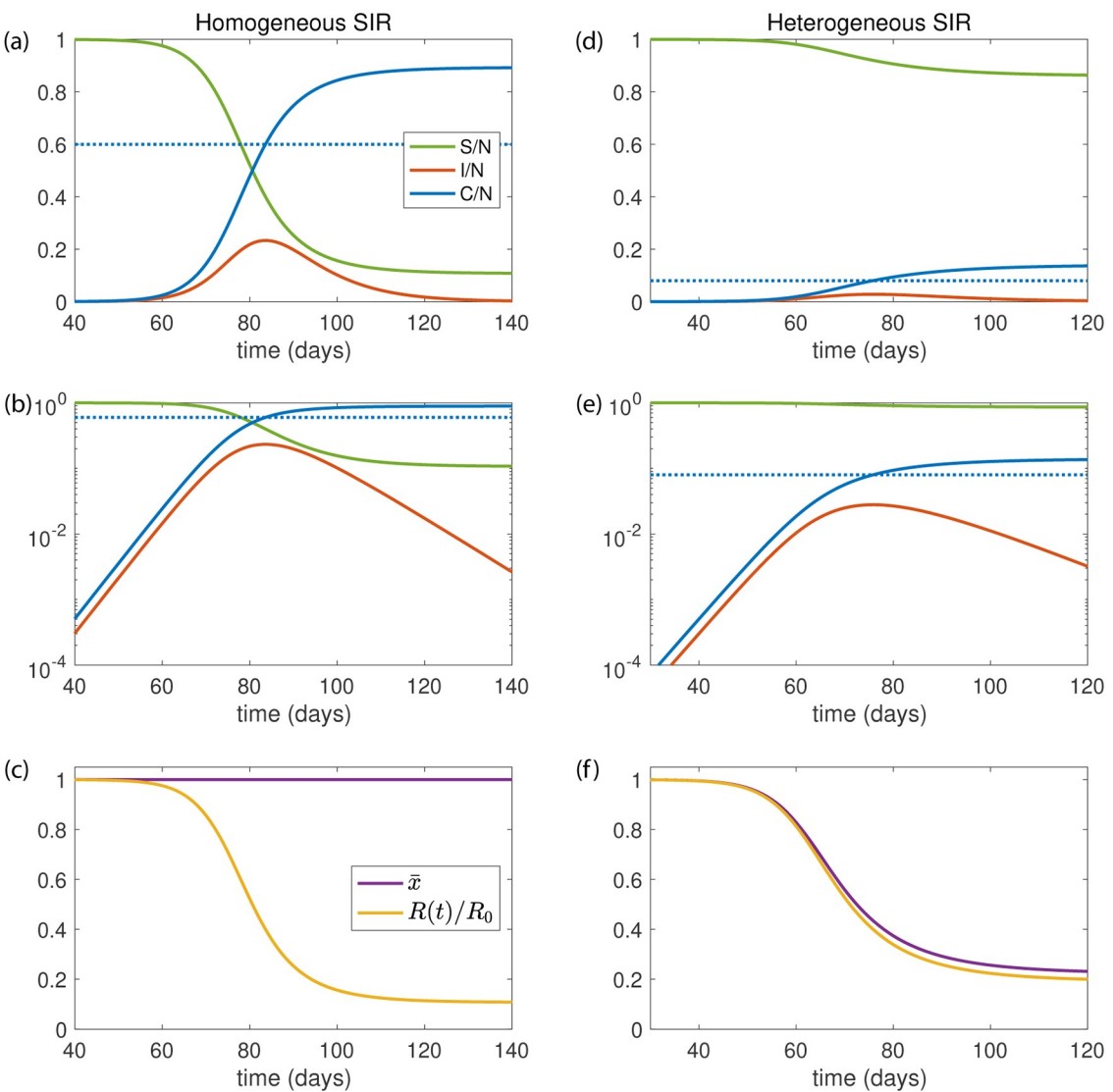

**Fig 1. Effects of population heterogeneity on the dynamics of SIR models.** Examples for the time course of fraction of susceptible $S/N$ (green), fraction of infected $I/N$ (orange) and fraction of the cumulative number of infected $C/N$ (blue) in a SIR model with $N$ total individuals. (a)-(c) Homogeneous SIR model with $R_0 = 2.5$ and $\gamma = 0.13$ day$^{-1}$. (d)-(f) Heterogeneous SIR model with same $R_0$ and $\gamma$ and with $\alpha = 0.1$. (a) and (d) show time course as linear plot, (b) and (e) show semi logarithmic plots of the same variables. (c) and (f) show the normalized time dependent reproduction number $R(t)/R_0$ (yellow) and the average susceptibility $\bar{x}(t)$ (purple) as a function of time. The dotted lines in (a),(b),(d) and (e) indicate the herd immunity level $C_I$. Other parameters: $N = 8 10^7$ individuals and $I_0 = 10$ initially infected.

classic SIR model. An example for a solution to these equations for homogeneous population $\bar{x} = 1$ and constant $\beta$ is given in Fig 1(a) and 1(b). The corresponding time dependent reproduction number is presented in Fig 1(c). The number of infections first grows exponentially with growth rate

$$\lambda_0 = \gamma(R_0 - 1) \quad . \tag{5}$$

As the number of susceptible decreases, the epidemic reaches a peak number of infected $I_{\max} = I(t_I)$ at time $t = t_I$ with $\dot{I}(t_I) = 0$ and $R(t_I) = 1$. At this peak, a fraction $S_I/N = 1/R_0$ of

individuals remain susceptible. The cumulative number of infections $C_I$ at the maximum of $I$ thus obeys

$$\frac{C_I}{N} = 1 - \frac{1}{R_0} \quad . \tag{6}$$

Eq (6) is the classic herd immunity level which is the fraction of immune individuals in the population beyond which the epidemic can no longer grow. Finally the epidemic dies down exponentially with rate

$$\lambda_\infty = \gamma \left( \frac{R_0 S_\infty}{N} - 1 \right) \quad , \tag{7}$$

where

$$\frac{S_\infty}{N} = -\frac{1}{R_0} W(-R_0 e^{-R_0}) \tag{8}$$

is the fraction of susceptible individuals that remain after long times, see Appendix A. Here $W(z)$ denotes Lambert W-function [28]. The total fraction of infections over the course of the epidemic is $C_\infty/N = 1 - S_\infty/N$.

For a classic SIR model with homogeneous population we have for $R_0 = 2.5$, a herd immunity level $C_I/N$ of 60% of the population, see Fig 1(a) and 1(b). After the infection has passed $C_\infty/N \simeq 89\%$ of the population have been infected, see Fig 2(a) and 2(b) (green lines). The fraction of the population that become infected increase for larger $R_0$. The SIR model thus suggests that for $R_0 > 2$ the epidemic wave exceeds a majority of the population before the epidemic begins to die out.

## B. Infection dynamics with population heterogeneity

Not all individuals are the same and for some susceptible individuals the probability of infection per time is lower than for others. This can be captured by a distribution of susceptibilities $x$ [13, 15, 16]. We denote $s(x)dx$ the number of individuals with susceptibility between $x$ and

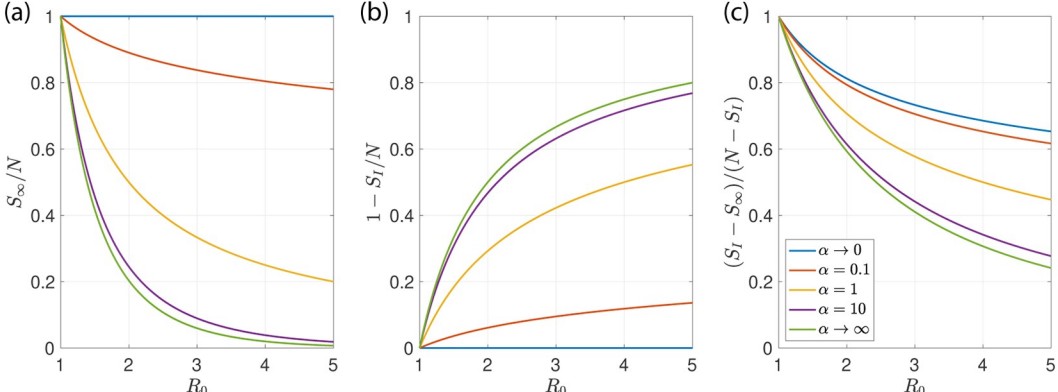

**Fig 2. Fraction of susceptible individuals at long times.** (a) Fraction $S_\infty/N$ of susceptible individuals that remain at long times as a function of the basic reproduction number $R_0$ for different degrees of population heterogeneity characterized by the values of $\alpha$. The limit $\alpha \to \infty$ corresponds to the classic case with homogeneous populations (green). In the limit $\alpha \to 0$ populations are most heterogeneous (blue). (b) Fraction $S_I/N$ of susceptible individuals as a function of $R_0$ at the peak where the number of infected is maximal for different $\alpha$. (c) Ratio of infections after the peak $S_m - S_\infty$ and infections before the peak $N - S_m$ as a function of $R_0$.

$x + dx$. The total number of susceptible individuals is then $S(t) = \int_0^\infty dx\, s(x, t)$. For each sub-population $s(x)$ with susceptibility $x$, the number of susceptible individuals decreases as

$$\partial_t s = -\beta x s \frac{I}{N} \quad , \tag{9}$$

which for the whole population implies Eq (1) with average susceptibility

$$\bar{x}(t) = \frac{1}{S(t)} \int_0^\infty dx\, x s(x, t) \quad , \tag{10}$$

which is in general time dependent.

This time dependence can be discussed by introducing the variable $\tau$ that increases monotonically during the epidemic wave. It starts with $\tau = 0$ and is defined via the equation $\dot{\tau} = \beta I / N$, which implies that it reaches a final value when the epidemic has decayed. Therefore $\tau$ can be interpreted as a measure of how far the epidemic has progressed. Eq (9) can then be written as $\partial_\tau s = -x s$, and the number of susceptible individuals is

$$S(\tau) = \int_0^\infty dx\, s_0(x) e^{-\tau x} \quad , \tag{11}$$

where $s_0(x)$ is the initial susceptibility distribution at time $t = t_0$ with average $\bar{x} = 1$, see Appendix B.

## C. Infection dynamics with generic power law heterogeneity

The dynamics of epidemic waves depends on the shape of the initial distribution $s_0(x)$. Here, we consider distributions that have the special property of shape invariance under the dynamics of epidemics. This property is satisfied by a gamma distribution

$$s_0(x) \sim x^{-1+\alpha} e^{-\alpha x} \quad , \tag{12}$$

which is governed by a power-law at small $x$, characterized by the exponent $\alpha > 0$, and a cut off at large $x$, see Eq (C1) in Appendix C. The distribution $s_0(x)$ has average $\bar{x} = 1$ and variance $1/\alpha$. Indeed, we have $s(x, t) = \bar{x}^{-1+\alpha} s_0(x/\bar{x})$, where the time dependence enters via $\bar{x}(t)$, see Appendix C. This shape invariance implies that the gamma distribution is maintained at all times and is not merely an initial condition. Furthermore, starting with any initial distribution that exhibits a power law $s_0(x) \sim x^{-1+\alpha}$ at small $x$, the distribution will converge for large $\tau$ to the shape invariant gamma distribution, which therefore is an attractor of the dynamics, see Appendix B. Note that in the limit of large $\alpha$, we recover the classic SIR model for a homogeneous population. For small $\alpha$, the population is strongly heterogeneous.

By inserting the distribution given in Eq (12) into Eq (11) we obtain

$$S(\tau) = \frac{N - I_0}{\left(1 + \frac{\tau}{\alpha}\right)^\alpha} \quad , \tag{13}$$

as detailed in Appendix C. The average susceptibility is

$$\bar{x} = \frac{1}{1 + \frac{\tau}{\alpha}} \quad , \tag{14}$$

which starts from $\bar{x} = 1$ for $\tau = 0$ and decreases for increasing $\tau$, thus dampening the epidemic. We can now express the dynamics given in Eqs (1) and (2) as two dynamic equations for $I(t)$

and $\tau(t)$ which read

$$\dot{I} = I\beta(1 - \frac{I_0}{N})(1 + \frac{\tau}{\alpha})^{-(\alpha+1)} - \gamma I \tag{15}$$

$$\dot{\tau} = \beta I/N \quad . \tag{16}$$

with initial values $I(0) = I_0$, $\tau(0) = 0$ and $S(0) = N - I_0$. The number of susceptible individuals at time $t$ is then given by $S(t) = S(\tau(t))$. An example of a time course of this model for $\alpha = 0.1$ is shown in Fig 1(d), 1(e) and 1(f).

We can discuss how the shape of the epidemic wave depends on the parameter $\alpha$. The epidemic starts out with exponential growth of infected individuals at rate $\lambda_0 = \gamma(R_0 - 1)$, with $R_0 = \beta/\gamma$. The time dependent reproduction number is

$$R = \bar{x}^{1+\alpha}R_0 \quad . \tag{17}$$

When the reproduction number drops to $R = 1$, the number of infected reaches a maximum

$$\frac{I_{\max}}{N} = 1 - \frac{1}{R_0} - \frac{1}{R_0}(1 + \alpha)\left(R_0^{\frac{1}{1+\alpha}} - 1\right) \quad . \tag{18}$$

Beyond the herd immunity level given by the cumulative number of infections at the maximum of $I$

$$\frac{C_I}{N} = 1 - R_0^{-\frac{\alpha}{\alpha+1}} \quad , \tag{19}$$

the reproduction number $R$ drops below 1 and the epidemic dies down. In Eqs (17)–(19) we have considered the limit of small $I_0/N$ for simplicity. The general derivation is given in Appendix C. In the limit of large $\alpha$, these expressions converge to those obtained for the homogeneous SIR model, see Appendix C. The remaining fraction of susceptible individuals at the peak and after the epidemic has passed is shown as a function of $\alpha$ in Fig 2(a) and 2(b). This reveals that as $\alpha$ is reduced, the fraction of the population reached by the epidemic decreases and can become very small for small $\alpha$. At the same time the infections are more spread out over time and a larger fraction occurs after the peak when $\alpha$ is reduced, see Fig 2(c).

An important case is a strongly heterogeneous population. For small $\alpha \ll 1$, we obtain simple analytical expressions for the behavior of the system, see Appendix E. In this limit we have $I_{\max}/N \simeq \alpha(\ln R_0 + 1/R_0 - 1)$ and $C_I/N \simeq \alpha \ln R_0$. An important quantity is the rate $J$ of new cases per time. For small $\alpha$ it takes the maximal value

$$\frac{J_{\max}}{N} \simeq \gamma\alpha((R_0 - 2)e^{\frac{1}{R_0}-1} + 1) \tag{20}$$

The final number of susceptible individuals is given by

$$\frac{S_\infty}{N} = \bar{x}_\infty^\alpha \quad , \tag{21}$$

where for small $\alpha$ the average susceptibility after the infection has passed is $\bar{x}_\infty \simeq -1/(R_0 W_{-1}(-e^{-1/R_0}/R_0))$. Here $W_{-1}(z)$ denotes the $-1$ branch of Lambert $W$ function.

We finally have for small $\alpha$

$$\lambda_\infty = \gamma(R_0 \bar{x}_\infty - 1) \quad . \tag{22}$$

A key result is that for small $\alpha$ the herd immunity level can be much below the classical value suggested by the SIR model. For example for $R_0 = 2.5$ and $\alpha = 0.1$, we have $I_{max}/N \simeq 2.8\%$, and a fraction $C_I/N \simeq 8\%$ of infected individuals required for herd immunity, much lower than is usually suggested. The total number of infected at long times is $C_\infty/N \simeq 14\%$, see Appendix C. The reason for the small amplitude of the epidemic wave in a heterogeneous population (see Fig 1(d) and 1(e)) as compared to the amplitude for a homogeneous population (see Fig 1(a) and 1(b)) is the drop of the average infection susceptibility $\bar{x}$ (see Fig 1(f)). This drop occurs because in a heterogeneous population the most susceptible individuals are first removed, thereby lowering the average susceptibility.

## III. Application to the SARS-CoV-2 epidemic in Germany

We analyze the dynamics of the SARS-CoV-2 epidemic in Germany using public data provided by the Robert Koch institute [26]. These daily reports provide the numbers of reported positive tests for each day, but also the dates of reporting of those infections which later turn out as fatal. The total number of new reported infections per day $J_{rep}^t$ (red symbols) are shown in Fig 3(a) together with the number of reported infections per day that were later fatal (blue symbols), which we denote $J_{rep}^f(t)$. Both sets of data can be interpreted as proxies for the rate $J$ of new cases per day up to an unknown factor. They show qualitatively similar behavior, a rapid growth and a decline after passing a maximum. However there are quantitative differences, in particular the growth rates at early and late times, given by the slopes of the data in a single logarithmic plot are different, see Fig 3(b). The number of new cases per day that are later fatal $J_{rep}^f(t)$ is related to the number of new infections per day as $J_{rep}^f(t) = Jf$, where $f$ denotes the infection fatality rate, the fraction of infections that are fatal, which we consider to be constant for simplicity.

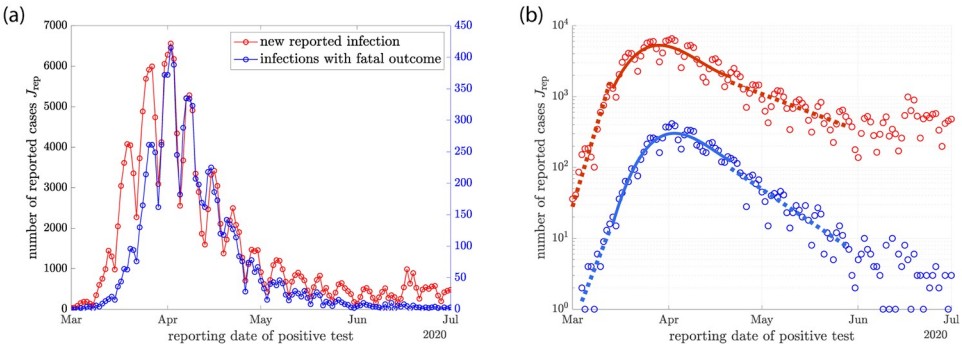

**Fig 3.** (a) Daily new SARS-CoV-2 infections reported in the early months of 2020 in Germany. The number of new reported infections per day $J_{rep}^t$ (red symbols) is shown together with the number of reported infections per day for those cases with later fatal outcome $J_{rep}^f$ (blue symbols). (b) Semi logarithmic representation of the same data. The dashed and solid lines represent linear and cubic fits to the data in specific time intervals. They are used to estimate the initial and final growth rates $\lambda_0$ and $\lambda_\infty$ as well as $A_2 = \ddot{J}/J$ and $A_3 = \dddot{J}/J$ at the maximum of the rate of new cases $J_{max}$. We find $\lambda_0 \simeq 0.269$ day$^{-1}$ (0.336 day$^{-1}$), $\lambda_\infty \simeq -0.068$ day$^{-1}$ ($-0.038$ day$^{-1}$), $A_2 \simeq -10^{-2}$ day$^{-2}$ ($-0.91 \, 10^{-2}$ day$^{-2}$) and $A_3 \simeq 6.8 \, 10^{-4}$ day$^{-3}$ ($7.5 \, 10^{-4}$ day$^{-3}$) for the fatal cases (for all reported cases).

## A. Comparison to heterogeneous SIR model

The calculated number of new cases per day $J$ obtained as solution to Eqs (15) and (16) for a heterogeneous population and scaled by the factor $f$ to match the data of fatal cases are shown in Fig 4(a)–4(d) as solid blue lines. These lines are shown together with the number $J_{rep}^f$ of new fatal cases per day as blue symbols. The factor $f$ was determined such that the cumulated cases per day $J_{rep}^f/f$ matches the cumulative number of cases $C$ on June 15. The time axis is chosen such that the model matches the data. From a fit of the model to the data we obtain the parameter estimates $R_0 \simeq 2.67$ and $\gamma \simeq 0.146$ day$^{-1}$. Good fits to the data are found for a range of $\alpha$ sufficiently small, about $\alpha < 0.2$. The resulting infection fatality rates $f$ vary as $\alpha$ is changed. Using $\alpha = 0.05$ corresponds to an infection fatality rate $f \simeq 0.13\%$. It could be larger or smaller if a different value of $\alpha$ was used. This would not significantly affect the quality of the fit as

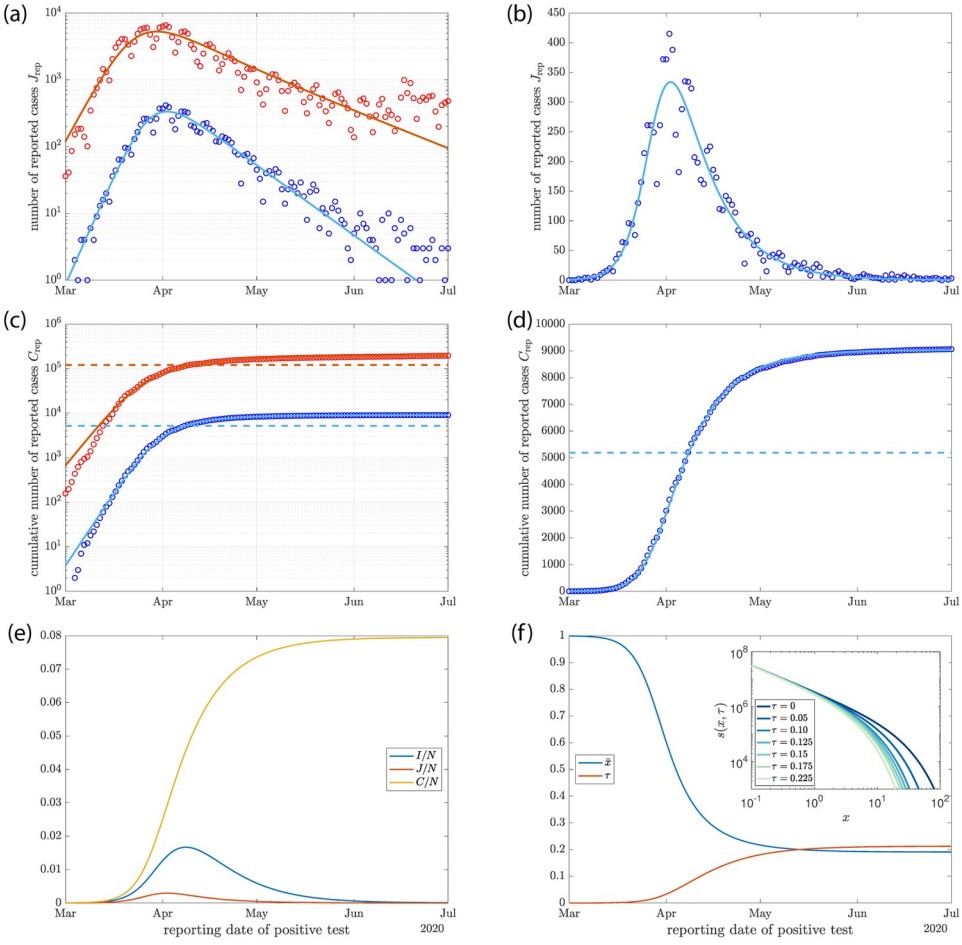

**Fig 4. Time course of the SARS-CoV-2 epidemic in Germany (symbols) compared to solutions of the heterogeneous SIR model (lines).** (a) and (b) Data on daily reported infections (red) and on reported infections with later fatal outcome (blue) as logarithmic and linear plots. (c) and (d) same data and model solutions as in (a) and (b) but for cumulative numbers of cases. The horizontal dashed lines indicate scaled herd immunity levels. Parameter values for the model solution are $R_0 = 2.67$ ($R_0 = 3.91$), $\gamma = 0.146$ ($\gamma = 0.069$), $\alpha = 0.05$ and $N = 8 \cdot 10^7$ for the fatal cases (for all cases). The case fatality rate that corresponds to this solution is $f = 0.13\%$ ($f = 0.11\%$). (e) Time courses of the fraction of infected $I/N$ (blue), the new cases per day $J/N$ (red) and the fraction of cumulative cases $C/N$ (yellow) for $R_0 = 2.67$ and $\gamma = 0.146$. (f) Time course of the average susceptibility $\bar{x} = (R/R_0)^{1/(1+\alpha)}$ (blue), where $R$ is the time dependent reproduction number and of $\tau = \alpha(1/\bar{x} - 1)$ (red) for the solution shown in (e). Inset: distributions of susceptibility in the population for different values of $\tau$.

long as $\alpha < 0.2$. The calculated time courses $I(t)$, $J(t)$ and $C(t)$ corresponding to these fits to $J^f_{\text{rep}}$ are shown in (e). The dependence of the average susceptibility $\bar{x}$ on time and the function $\tau(t)$ are shown in (f). The increase of $\tau$ with time represents the advance of the epidemic. It reaches a final value at long times as discussed in Appendix C. The inset in (f) shows the shape of the distribution of susceptibility in the population at different stages characterized by different values of $\tau$.

It is surprising that the model fits the data of fatal cases with just two fit parameters while yielding a reasonable infection fatality rate. This is further clarified when using the fit values of $R_0$ and $\gamma$ to calculate $\lambda_0 \simeq 0.24$ day$^{-1}$, slightly smaller than the estimate given in Fig 3(b). Using Eq (22), we also find $\lambda_\infty \simeq -0.069$ day$^{-1}$, very close to the estimate from the data. The data of all reported cases can also be captured by the model for small $\alpha$, see Fig 4(a) and 4(c) red symbols and red lines. This fit is not as close and the parameter values are different, see Fig 4. Our comparison of the model to the data shows that the model captures the time course of fatal cases surprisingly well for the case of strong heterogeneity for infection fatality rates that fall in the range of estimates from immunological studies [29–33].

## B. Quantification of the shape of the epidemic wave

In order to understand how the shape of the wave of infections constrains the possible parameter values of $R_0$, $\gamma$ and $\alpha$, we consider in addition to the initial growth rate $\lambda_0$ and the final decay rate $\lambda_\infty$ two coefficients describing the epidemic dynamics near its peak, using the expansion

$$\ln J(t) \simeq \ln J_{\text{max}} + \frac{A_2}{2}(t - t_J)^2 + \frac{A_3}{6}(t - t_J)^3 \quad , \tag{23}$$

where the linear term disappears by definition at the maximum $J_{\text{max}} = J(t_J)$ at time $t_J$. The coefficients $A_2 = \ddot{J}/J|_{t=t_J}$ and $A_3 = \dddot{J}/J|_{t=t_J}$ can be obtained for the homogeneous and heterogeneous SIR model, see Appendix C, D and E. Fig 5 shows dimensionless combinations of these values as a function of $R_0$ for different $\alpha$ ranging from the homogeneous case $\alpha \to \infty$ to strongly heterogeneous with $\alpha \to 0$ as solid lines of different color. The values obtained from the fits shown in Fig 3 are indicated as dashed lines together with shaded regions corresponding to estimated uncertainty ranges of these values.

We find that the ratio $A_2/\lambda_\infty^2$, which is independent of $\gamma$ depends only weakly on $\alpha$. We can therefore use it to estimate $R_0$, see Fig 5. Using $A_2 \simeq -0.01$ day$^{-2}$ and $\lambda_\infty \simeq -0.07$ day$^{-1}$ determined from the data of fatal cases, we have $A_2/\lambda_\infty^2 \simeq 2.0$ leading to the estimate $R_0 \simeq 2.5$, see Fig 5. This estimate can now be used to infer bounds on $\alpha$. The ratio $\lambda_\infty/\lambda_0 \simeq -0.3$ is only consistent with $R_0 \geq 2.6$ and the lower value corresponds to the limit of small $\alpha$, see Fig 5. This reveals that $\alpha \ll 1$ must be small and that the classic SIR model with homogeneous population is not consistent with this data. We can now estimate $\gamma$ using the small $\alpha$ limit. For $R_0 \simeq 2.6$, we have $\gamma \simeq 2\lambda_\infty \simeq 0.14$ day$^{-1}$, see Fig 5(d). The data does not provide information about the true total number of infections. Therefore the precise value of $\alpha$ remains unknown. We can use estimates from immunological studies estimating the number of infections [29, 31, 32] to determine $\alpha$. This suggests a range of about $0.01 < \alpha < 0.15$, corresponding to $0.65\% > f > 0.04\%$. Fig 5 also shows the estimated ranges for data on all reported cases in red. For this case the inferred values of $R_0$ is larger and the consistency with the data is less strong.

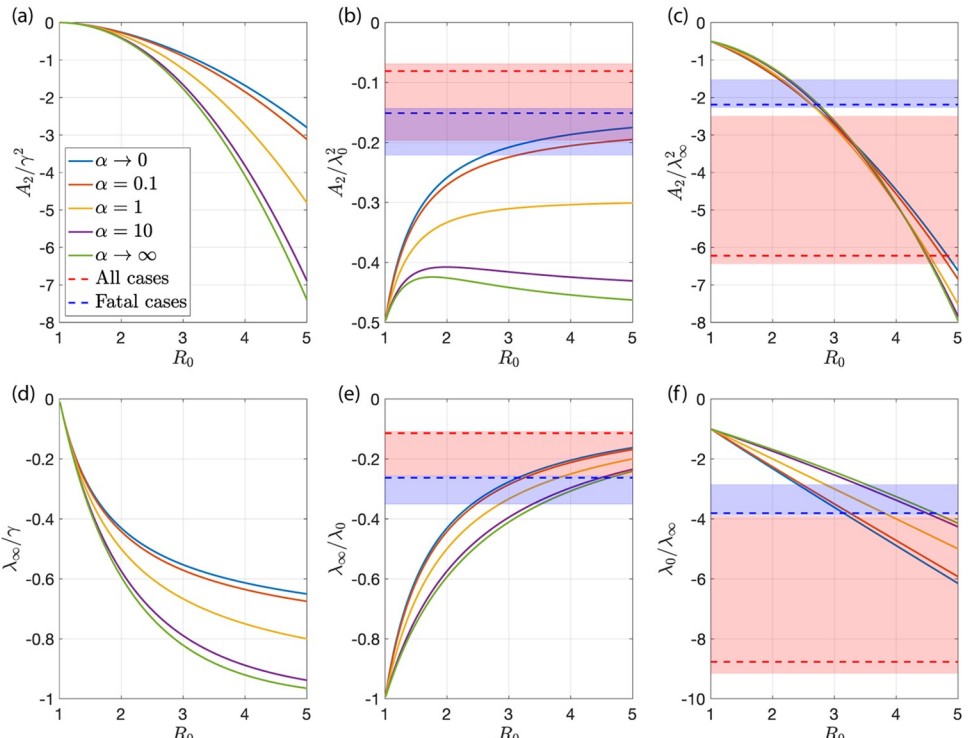

**Fig 5. Role of population heterogeneity for the behavior of the generalized SIR model.** Plots of various dimensionless ratios of parameters characterizing the shape of the infection wave for different values of $\alpha$. Here the limit $\alpha \to \infty$ corresponds to the homogeneous SIR model, the limit $\alpha \to 0$ to the strongly heterogeneous case. Here $\lambda_0$ and $\lambda_\infty$ denote the initial and final growth rate, $A_2 = \ddot{J}/J$ and $A_3 = \dddot{J}/J$ describe the shape of the wave at the maximum of new cases per day $J$. The horizontal dashed lines correspond to estimates from fits shown in Fig 3, the shaded regions indicate uncertainty ranges, see Appendix D.

## C. Effects of mitigation and social distancing measures

During an epidemic conditions can change over time. For example, mitigation by social distancing measures, quarantining or seasonal changes could affect how quickly an infection spreads on average from person to person. Given that such changes are global, they may be captured by a time-dependence of the rate $\beta(t)$ [10, 12, 27]. In the following, we discuss scenarios of mitigated epidemics, starting from a reference point with an initial infection rate $\beta_0$ prior to mitigation. We use this reference to define the herd immunity $C_I$ of the population via Eq (19). The herd immunity level depends on the basic reproduction number $R_0 = \beta_0/\gamma$ and on the population heterogeneity $\alpha$. For immunity levels above herd immunity, $C \geq C_I$, the population is stable after mitigation measures are completely relaxed and $\beta$ is restored to its original value $\beta_0$.

We examine three different scenarios with a comparable total number of infections. These scenarios are shown in Fig 6. They are characterized by different levels of immunity relative to herd immunity at July 1 and thus differ in the future behaviors beyond this time. Starting in all scenarios with $R_0 = 2$ and using $\gamma = 0.24$, the model follows the initial growth at rate $\lambda_0$ of the reported cases. If $\beta$ is kept constant, $\beta = \beta_0$, the model deviates from the data at later times, see dotted lines in Fig 6. If $\beta$ is permitted to change in time, almost any reported time course could be described by the model. We use the data to infer a time course of $\beta(t)$ such that the model follows the data, see Appendix G. The inferred values of $\beta$ are shown as circles in Fig 6(c), 6(f) and 6(i). In order to fit the model to the data in different mitigation scenarios, we use a

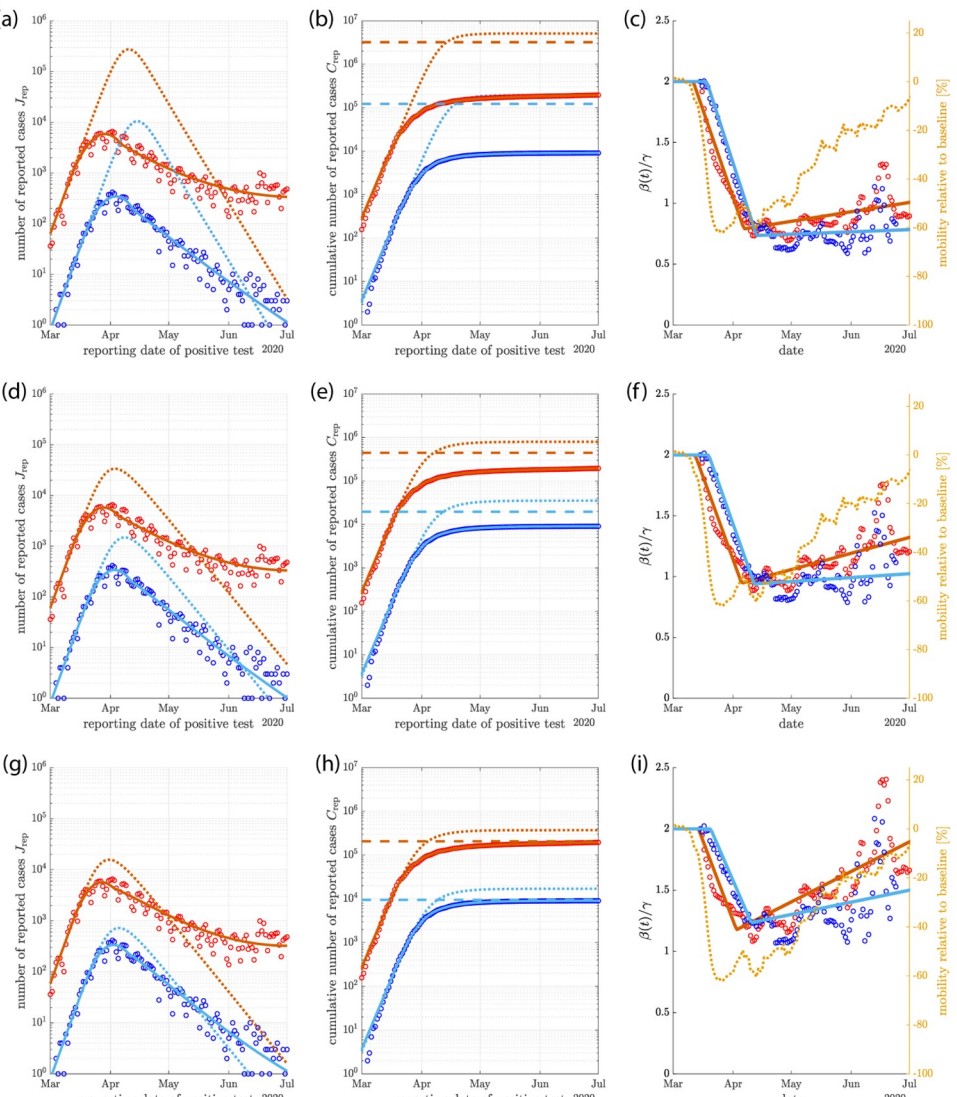

**Fig 6. Scenarios of mitigation.** (a)-(c) Early mitigation by strong reduction of $\beta$ for a homogeneous population (large $\alpha$ limit). The new cases per day are shown in (a) as symbols. A fit of the mitigated model is shown as solid lines. The solution for same parameter values $R_0 = 2$ and $\gamma = 0.24$ but without mitigation is shown as dotted lines. The corresponding cumulative numbers of cases are shown in (b). Herd immunity levels corresponding to these solutions are indicated as horizontal dashes lines. The time dependence of $\beta(t)$ are shown in (c) as solid lines. The time courses of $\beta$ inferred from the data is shown as symbols. Mobility data indicating social activities in Germany relative to baseline values are shown in orange for comparison. (d)-(f) same plots as in (a)-(c) but for a moderate mitigation and heterogeneous population with $R_0 = 2$, $\gamma = 0.24$ and $\alpha = 0.1$. (g)-(i) Heterogeneous population with mild mitigation and release leading to almost herd immunity. Red symbols and lines correspond to the case of all reported infections, blue data and lines correspond to reported infections of fatal cases.

piecewise linear modulation of $\beta$. The time dependence of $\beta(t)$ that resulted from these fits are shown in Fig 6(c), 6(f) and 6(i) as solid lines. The value of $\beta$ decreases sharply at the onset of mitigation. After this decrease it stays roughly constant or increases at constant rate, thus relaxing mitigation. The magnitude of maximal mitigation and the two slopes of $\beta(t)$ were used as fit parameters.

In the case of early mitigation, Fig 6(a)–6(c), fast reduction of $\beta$ suppresses the epidemic before any appreciable progress towards herd immunity was made. Mitigation needs to be

strong and sustained to be compatible with the data. By July, the population reaches only about 6–7% of herd immunity in this case. Note that this is the only scenario of the classic SIR model with a homogeneous population ($\alpha \to \infty$) that could be compatible with the data.

For heterogeneous populations with $\alpha \lesssim 0.2$, scenarios with milder mitigation and with infection levels closer to herd immunity are compatible with the data, see Fig 6d and 6g. A case of moderate mitigation with $\alpha = 0.1$ is shown in Fig 6d–6f. The population in this case reaches by July 1st about ~45% of herd immunity. A sustained mitigation is needed to account for the data, albeit with smaller magnitude compared to the first case. If the epidemic starts slightly earlier (3 days for the case shown in Fig 6g–6i as compared to (d-f)), the population reaches ~95% of herd immunity by July 1st. Here, mitigation has the effect to reduce the cumulative number of infections as compared to a non-mitigated case ($C/N = 5.8\%$ compared to 11% by July 1st). This reduction of cumulative infections $C$ results from a reduction of the number of infectious individuals $I$ at the point when herd immunity is reached. In the absence of mitigation, $I$ reaches its maximum when $C = C_I$, whereas mitigation can reduce $I$ to small numbers as herd immunity is reached, preventing further infections. The minimal number of infections that can be achieved by temporary mitigation is $C_I$, which is up to 50% smaller than the long time limit $C_\infty$ in an unmitigated epidemic (see Fig 2c).

The scenarios of temporary reduction of $\beta$ could capture the mitigation effects of social distancing measures. To relate the inferred time dependence of $\beta$ to measures of social activity, we show in Fig 6(c), 6(f) and 6(i), $\beta(t)$ together with mobility data from Ref. [34] for comparison, see Appendix H. This mobility data shows a sharp decline and a slow but steady return to the initial state roughly in line with inferred changes of $\beta(t)$.

The three scenarios differ in the fraction of herd immunity they reach by July 1 and therefore in their future trajectories. However, $\beta(t)$ was adjusted by a fitting procedure such that all scenarios are consistent with the data on reported infections. This reveals that it can be difficult to distinguish effects of heterogeneity leading to a time dependent average susceptibility $\bar{x}$ from mitigation effects corresponding to time-dependent $\beta$. Indeed our analysis shows that changes in mitigation strength can be compensated to some degree by changes of heterogeneity described by $\alpha$.

## IV. Conclusions and perspectives

We have presented a generalization of the classic Susceptible-Infected-Removed model for epidemic waves, which adds one new parameter to the model that captures population heterogeneity by a power-law exponent $\alpha$. This exponent describes the power law that characterizes the distribution of susceptibility in the population $s(x) \sim x^{-1+\alpha}$ for small $x$. A special case for such distributions is the gamma distribution. Gamma distributions have been used before to describe heterogeneous populations [13, 15–17] and an approach similar to the one presented here has been proposed in [15] where also the shape invariance of gamma distributions is mentioned. Here, we have shown that gamma distributions have the special properties that they are both shape invariant under the dynamics and attractors of the dynamics for power-law distributions. This implies that for each $\alpha$ there exists a class of distributions with the same power law at small $x$ which share the same limiting dynamics and distribution. The generalization of the SIR model introduced here captures the effects of these power laws by the parameter $\alpha$ in a generic way. Note that this generalization does not change the simplistic nature of the SIR model and does not change its numerical or analytical complexity.

For $\alpha > 1$, population heterogeneity is weak and in the limit of large $\alpha$, one recovers the classic SIR model of homogeneous population, see Fig 1(a)–1(c). For $\alpha < 1$ population heterogeneity plays a key role in limiting the peak of the epidemic wave. We show that as a result of

strong population heterogeneity (small $\alpha$), the wave peaks when only a small minority of individuals have been infected, see Fig 1(d)–1(f). The herd immunity level, the point where the epidemic dies down spontaneously becomes very small for small $\alpha$, see Eq (19). Thus our model shows that for small $\alpha$, an epidemic wave can die out after reaching only a small fraction of the population even though a majority of the population is still susceptible. In this case the population is stable with respect to introducing new infected individuals because the average susceptibility $\bar{x}$ has dropped significantly, see Fig 1(f).

Many properties of the nonlinear wave in this generalized model can be obtained exactly as a function of $\alpha$ and in the limit of small $\alpha$. Numerical solutions can be generated quickly and efficiently. In a heterogeneous population the average susceptibility $\bar{x}$ stays almost constant at early stages of the epidemic where the number of new cases grows exponentially with rate $\lambda_0 = \gamma(R_0 - 1)$. At this stage the dynamics is the same as in the classic model and independent of $\alpha$. However, $\bar{x}$ then drops rather quickly and the epidemic waves thus reaches its peak and dies down, see Fig 1(b) and 1(f). This sudden drop in average susceptibility results from a shift of the distribution of susceptibility. The most susceptible individuals are removed from the dynamics at higher rates than those with low susceptibility. This leads to a rapid reduction of the average susceptibility until it has dropped to a low value where the time dependent reproduction number $R$ falls below 1, see Eq (17). The wave then dies down at rate $\lambda_\infty$ and the average susceptibility approaches a final value $\bar{x}_\infty$. Thus the qualitative behavior of the classic SIR model is unchanged and the key parameters, the recovery rate $\gamma$ and the basic reproduction number $R_0$ have the same values and properties. However, the power-law distribution of susceptibility can dramatically change the peak of the epidemic and alters the precise shape of the wave. The dynamics effectively shifts the edge of the susceptibility distribution, see inset in Fig 4(f), which changes the stability of the population from prone to an exponentially growing wave to a stably decaying wave without requiring a large number of infections.

The simple SIR model does not aim to capture details such as the population structure, the geography or human travel. In the spirit of statistical physics it is based on the idea that the collective behaviors of many individuals give rise to an emergent epidemic wave with robust and generic features that can be captured by a simplified model that focuses on key aspects. Here we show that power-law heterogeneity is a key factor that should not be left out. Note that our approach to take into account population heterogeneity can also be applied to extensions and generalizations of the SIR model such as the SEIR and the SIRS models [16, 35–37].

We apply our model to data on the SARS-CoV-2 epidemic in Germany in 2020. The data from Germany provides time stamps on reporting dates of infections and reporting dates of infections that later are fatal. Surprisingly, the data for fatal cases is well described by the heterogeneous SIR model with constant parameters and small $\alpha$ but not by the classic SIR model with constant parameters. In the case of SARS-CoV-2, immunological data suggests that only a minority of the population exhibits antibodies [29, 31, 32]. This is consistent with a fit of our model to the data using a small value of $\alpha$. The data on all reported cases can also be captured by the model, but the fit is less convincing. Comparing the data on all reported cases to the data on the time course of cases that are fatal reveals some differences. Clearly the fatal cases represent a different sampling as these cases correspond to predominantly old individuals and therefore measure a different quantity. However, starting from all reported cases and then using the fatal outcome as a second criterion could reduce biases due to changes in testing rates, testing strategies as well as testing errors.

An epidemic wave does not progress under constant conditions but is subject to changes such as mitigation measures and seasonal effects. We use our model in comparison with the data from Germany to investigate different scenarios of mitigation that correspond to different level of immunity in the population. In the case of a homogeneous population the data can

only be accounted for if mitigation is strong and suppresses the epidemic far below herd immunity Fig 6(a)–6(c). This scenario further requires mitigation to be sustained and it leads to a fragile and unstable state when mitigation measures are relaxed.

In the case of heterogeneous populations, intermediate scenarios are possible which stay below herd immunity or just reach herd immunity, see Fig 6(d)–6(i). In the latter example shown in Fig 6(g)–6(i), mitigation effectively reduces the total number of infections by keeping immunity just at herd immunity level, leading to a stable state where mitigation can be safely relaxed. This is a desirable outcome because the number of infections could be reduced by mitigation by up to 40% without the need of sustaining mitigation, see Fig 2(c).

When discussing rapidly evolving epidemics such as SARS-CoV-2, herd immunity is often not considered to be reachable as it is predicted to require an unacceptably high fraction of cumulative infections [38]. Interestingly, the picture changes dramatically if a strongly heterogeneous population is considered. In this case herd immunity can be reached rather quickly while a large majority of the population is still susceptible. This raises the question of what are the features that are variable and that give rise to heterogeneity and how widely they are expected to vary in the population. One possibility is that differences of susceptibility stem from differences in the abilities of immune systems of susceptible individuals to react to a new pathogen. In addition to adaptive immunity related to the presence of specific antibodies, many individuals show a T-cell response to SARS-CoV-2 [25]. This response could for example be due to less specific cross reactions related to earlier encounters with related viruses [24].

Here we have focused on data from Germany until July 2020 because it provides detailed information that is not available in most countries. Furthermore, Germany has relatively few reported infections and deaths per capita. Our work shows that even this rather mild manifestation of the epidemic can be captured by a heterogeneous SIR model with mild mitigation. The data of newly infected cases with fatal outcome can even be explained in a strongly heterogeneous population without considering any mitigation effects at all. This implies that in order to quantify the effects of mitigation, population heterogeneity has to be taken into account. In order to disentangle effects from heterogeneity and from mitigation the combination of different types of information is important. For example, analyzing in different countries the circumstances under which different sero-prevalence levels or multiple epidemic waves are observed could be key to understand the roles of mitigation and heterogeneity.

## Appendix A: Properties of the homogeneous SIR model

For a homogeneous population with $\bar{x} = 1$, the SIR model given in Eqs (1 and 2) can be written as

$$\dot{I} = (1 - \frac{I_0}{N})\beta I e^{-\tau} - \gamma I \tag{A1}$$

$$\dot{\tau} = \beta \frac{I}{N} \tag{A2}$$

with $S = (N - I_0)\exp(-\tau)$, $I(0) = I_0$ and $\tau(0) = 0$. We therefore have $dI/d\tau = \dot{I}/\dot{\tau} = (N - I_0)e^{-\tau} - N/R_0$, where $R_0 = \beta/\gamma$. For constant $\beta$ this implies

$$I(\tau) = I_0 e^{-\tau} + N(1 - e^{-\tau}) - \frac{N\tau}{R_0} \quad . \tag{A3}$$

We can eliminate $\tau$ and find

$$\frac{I}{N} = 1 - \frac{S}{N} + \frac{1}{R_0} \ln \frac{S}{N - I_0} \quad . \tag{A4}$$

The maximum $I_{max} = I(\tau_I)$ with $I'(\tau_I) = 0$, where the prime denotes a $\tau$-derivative, occurs for

$$\tau_I = \ln \left( R_0 (1 - \frac{I_0}{N}) \right) \quad . \tag{A5}$$

Therefore we the maximal number of infected individuals reads

$$I_{max} = N \left[ 1 - \frac{1}{R_0} - \frac{1}{R_0} \ln \left( R_0 (1 - \frac{I_0}{N}) \right) \right] \quad . \tag{A6}$$

At the maximum $I_{max}$, we have $dI/dS = 0$, which implies

$$\frac{S(\tau_I)}{N} = \frac{1}{R_0} \quad . \tag{A7}$$

At long times, the infection dies out when $I(\tau_\infty) = 0$ with $(1 - I_0/N)\exp(-\tau_\infty) = 1 - \tau_\infty/R_0$ and $S_\infty = (N - I_0)\exp(-\tau_\infty)$. We therefore have $S_\infty/N = 1 + \ln(S_\infty/(N - I_0))/R_0$ and

$$\frac{S_\infty}{N} = -\frac{1}{R_0} W \left( -R_0 e^{-R_0} (1 - \frac{I_0}{N}) \right) \quad , \tag{A8}$$

where $W(z)$ denotes the 0-branch of Lambert W-function. The time dependent solution $\tau(t)$ can be obtained from $Nd\tau/I(\tau) = \beta dt$ via

$$\int_0^\tau \frac{d\tau'}{1 - (1 - I_0/N)e^{-\tau'} - \tau'/R_0} = \beta t \quad . \tag{A9}$$

To discuss empirical data, we consider the time-course of the rate of new cases $J = \beta IS/N$. We have $\dot{J}/J = \beta K$ with

$$K = (1 - \frac{I_0}{N})e^{-\tau} - \frac{1}{R_0} - \frac{I}{N} \quad . \tag{A10}$$

The maximum $J_{max} = J(\tau_J)$ is reached for $K(\tau_J) = 0$, which implies

$$\tau_J = W_{-1} \left( -(1 - \frac{I_0}{N})2R_0 e^{-(R_0+1)} \right) + R_0 + 1 \quad , \tag{A11}$$

where $W_{-1}(z)$ denotes the $-1$ branch of the Lambert $W$-function with $W(z)e^{W(z)} = z$ [28]. At the maximum $J_{max}$ of $J$ we have $\dot{S}I + \dot{I}S = 0$ and therefore

$$S(\tau_J) = -\frac{1}{2R_0} W_{-1}(-2R_0 e^{-1-R_0}) \tag{A12}$$

$$I(\tau_J) = S(\tau_J) - \frac{1}{R_0} \tag{A13}$$

and finally

$$J_{\max} = \beta S(\tau_J)\left(\frac{S(\tau_J)}{N} - \frac{1}{R_0}\right) \quad . \tag{A14}$$

Near the maximum of the rate $j_{\max}$ with $\dot{J} = 0$ we have $A_2 = \ddot{J}/J|_{t=t_J}$ and $A_3 = \dddot{J}/J|_{t=t_J}$. For the homogeneous SIR model, we have

$$A_2 = -\frac{\gamma^2}{2}[1 + W_{-1}(-2R_0 e^{-1-R0})][2 + W_{-1}(-2R_0 e^{-1-R0})] \tag{A15}$$

$$A_3 = \frac{\gamma^3}{4}[2 + W_{-1}(-2R_0 e^{-1-R0})]^2 \quad . \tag{A16}$$

## Appendix B: Distributions of infection susceptibility in the population

For an initial distribution $s_0(x)$ of susceptible individuals with susceptibility $x$, we define $S(\tau) = \int_0^\infty dx s_0(x) e^{-\tau x}$. We can then write the dynamics of the epidemic spreading given in Eqs (1) and (2) as two equations for $I(t)$ and $\tau(t)$

$$\dot{I} = -\beta \frac{I}{N} \frac{dS}{d\tau} - \gamma I \tag{B1}$$

$$\dot{\tau} = \beta \frac{I}{N} \tag{B2}$$

with initial values $I(0) = I_0$, $\tau(0) = 0$ and $S(0) = N - I_0$. The number of susceptible at time $t$ is then given by $S(t) = S(\tau(t))$. Defining the cumulant-generating function $\Gamma(\tau) = -\ln S(\tau)$, we have

$$\bar{x} = \frac{d}{d\tau}\Gamma \tag{B3}$$

and the nth cumulant of $x$ is for $n > 1$ given by

$$\langle x^n \rangle_c = (-1)^{n+1} \frac{d^n}{d\tau^n}\Gamma \quad . \tag{B4}$$

The classic case with homogeneous population then corresponds to $s_0(x) = (N - I_0)\delta(x)$, see Appendix A.

Here we consider distributions which exhibit a power-law behavior for small $x$ with $s_0 \sim x^{-1+\alpha}$. For $\alpha > 0$ the power law must be cut off at large $x > x_0$ for the distribution to be normalizable. From $\partial_\tau s = -xs$, we have $s(x, \tau) = s_0(x)e^{-\tau x}$ which for $\tau \gg 1/x_0$ approaches $s(x, \tau) \sim x^{-1+\alpha} e^{-\tau x}$. The moments of this distribution can be obtained from the cumulant generating function $\Gamma(\tau) = -\ln \int_0^\infty dx x^{-1+\alpha} e^{-\tau x} = -\alpha \ln \tau + \text{const}$. The cumulants of this distribution are $\langle x^n \rangle_c = \alpha \tau^{-n}(n-1)!$. Using $\bar{x} = \alpha/\tau$, the susceptibility thus approaches for large $\tau$ the limiting distribution

$$s(x, \tau) \sim x^{-1+\alpha} e^{-\alpha x/\bar{x}} \quad , \tag{B5}$$

which is the gamma distribution.

## Appendix C: The generalized SIR model with population heterogeneity

We have shown in Appendix B that for susceptibility distribution with a power law at small $x$ the gamma distribution is an attractor of the dynamics. We therefore choose at time $t = 0$ a gamma distribution with average $\bar{x} = 1$ as initial condition. It is given by

$$s_0(x) = (N - I_0)\frac{\alpha^{\alpha}}{(\alpha - 1)!}x^{-1+\alpha}e^{-\alpha x} \quad . \tag{C1}$$

Here $(\alpha - 1)!$ denotes Euler's gamma function. Note that in the limit of large $\alpha$, this approaches the homogeneous case with $s_0(x) \simeq \delta(x - 1)$. We then have $S(\tau) = (N - I_0)(1 + \tau/\alpha)^{-\alpha}$ and $S' = -(N - I_0)(1 + \tau/\alpha)^{-(1+\alpha)}$, where the prime denotes a derivative with respect to $\tau$. As time evolves, the shape of the distribution $s(x, t)$ is time independent. Indeed, with $\partial_\tau s = -xs$ we have $s(x, \tau) = s_0(x)e^{-\tau x}$ and thus

$$s(x, \tau) = (N - I_0)\bar{x}^{-1+\alpha}f(x/\bar{x}) \tag{C2}$$

with $\bar{x}(\tau) = (1 + \tau/\alpha)^{-1}$. The time-invariant distribution is then given by

$$f(z) = \frac{\alpha^{\alpha}}{(\alpha - 1)!}z^{-1+\alpha}e^{-\alpha z} \quad . \tag{C3}$$

The dynamic equation of the heterogeneous SIR model read

$$\dot{I} = I\beta(1 - \frac{I_0}{N})(1 + \frac{\tau}{\alpha})^{-(\alpha+1)} - \gamma I \tag{C4}$$

$$\dot{\tau} = \beta I/N \quad . \tag{C5}$$

Defining $I' = \dot{I}/\dot{\tau}$, we have

$$I' = (N - I_0)(1 + \frac{\tau}{\alpha})^{-(\alpha+1)} - \frac{N}{R_0} \quad . \tag{C6}$$

For constant $\beta$, we then have by integrating over $\tau$

$$I = I_0(1 + \frac{\tau}{\alpha})^{-\alpha} + N(1 - (1 + \frac{\tau}{\alpha})^{-\alpha}) - \frac{N\tau}{R_0} \quad . \tag{C7}$$

The maximum of $I$ is reached for $\tau = \tau_I$ with $I' = 0$ and thus

$$(1 + \frac{\tau_I}{\alpha})^{\alpha+1} = (1 - \frac{I_0}{N})R_0 \quad . \tag{C8}$$

We thus obtain

$$\frac{I_{\max}}{N} = 1 - \frac{1}{R_0} - \frac{1}{R_0}(1 + \alpha)[((1 - \frac{I_0}{N})R_0)^{\frac{1}{1+\alpha}} - 1] \quad . \tag{C9}$$

The herd immunity level $C_I/N$ is obtained by using $C_I = N - S_I$, where $S_I$ is derived using Eq (13) together with Eq (C8). It follows that

$$\frac{C_I}{N} = 1 - \frac{N - I_0}{N}((1 - \frac{I_0}{N})R_0)^{-\frac{\alpha}{\alpha+1}} \quad . \tag{C10}$$

The epidemic ends at long times for $I(\tau_\infty) = 0$, for which

$$(1 - \frac{I_0}{N})(1 + \frac{\tau_\infty}{\alpha})^{-\alpha} = 1 - \frac{\tau_\infty}{R_0} \tag{C11}$$

with $S_\infty$ individuals that remain susceptible. This quantity obeys

$$R_0 \frac{S_\infty}{N} + \alpha(1 - \frac{I_0}{N})^{\frac{1}{2}}(\frac{S_\infty}{N})^{-\frac{1}{2}} = R_0 + \alpha \quad . \tag{C12}$$

We then find

$$\frac{S_\infty}{N} = \frac{R_0 + \alpha}{R_0} F\left(\frac{\alpha}{R_0 + \alpha}\left[\frac{R_0\left(1 - \frac{I_0}{N}\right)}{R_0 + \alpha}\right]^{\frac{1}{\alpha}}, \frac{1}{\alpha}\right) \quad , \tag{C13}$$

Where the function $F(z, v)$ is defined as the inverse of the function $x^v(1 - x)$ via the condition $F^v(1 - F) = z$. Finally, using $Nd\tau/I(\tau) = \beta dt$, the time dependent solution $\tau(t)$ can be written as

$$\int_0^\tau \frac{d\tau'}{1 - (1 - I_0/N)(1 + \tau'/\alpha)^{-\alpha} - \tau'/R_0} = \beta t \quad . \tag{C14}$$

## Appendix D: Dynamics of the rate of daily new cases

Data on the dynamics of the epidemic typically provides information about new cases reported per day. We therefore consider the rate of nex cases $J = -\beta I S'/N$, where the prime denotes a $\tau$ derivative. Using $I' = -S' - N/R_0$, we have

$$I(\tau) = I_0 + S(0) - S(\tau) - \frac{N\tau}{R_0} \quad . \tag{D1}$$

We then write

$$\frac{\dot{J}}{J} = \beta K \tag{D2}$$

with

$$K = -\frac{S'}{N} - \frac{1}{R_0} + \frac{I}{N}\frac{S''}{S'} \quad . \tag{D2}$$

We then have

$$K' = -\frac{S''}{N} + \frac{I'}{N}\frac{S''}{S'} + \frac{I}{N}\frac{S'''S' - S''^2}{S'^2} \tag{D4}$$

The maximum of $J$ occurs at $\tau = \tau_J$ with $K(\tau_J) = 0$. We thus have $A_2 = (\ddot{J}/J)|_{t=t_j} = \beta\dot{K}$ and $A_3 = J^{-1}(d^3J/dt)|_{t=t_j} = \beta\ddot{K}$.

Plugging in $S(\tau) = N(1 + \tau/\alpha)^\alpha$ for the heterogeneous SIR model with $I_0 \ll N$, yields

$$K = \left(1 + \frac{\tau}{\alpha}\right)^{-1}\left(\left(1 + \frac{\tau}{\alpha}\right)^{-\alpha} - \frac{\alpha + 1}{\alpha}\frac{I}{N}\right) - \frac{1}{R_0} \tag{D5}$$

$$= \frac{1}{\alpha + \tau}\left(\left(1 + \frac{\tau}{\alpha}\right)^{-\alpha}(2\alpha + 1) - (\alpha + 1)(1 - \frac{\tau}{R_0})\right) - \frac{1}{R_0} \tag{D6}$$

At the maximum in $J$, we have $K = 0$, yielding

$$\frac{2\alpha + 1}{(\alpha + 1)\left(\frac{\alpha}{R_0} + 1\right)} = \left(1 + \frac{\tau_J}{\alpha}\right)^\alpha\left[1 - \frac{\alpha^2}{(\alpha + 1)(\alpha + R_0)}\left(1 + \frac{\tau_J}{\alpha}\right)\right] \tag{D7}$$

Using the function $F(z, v)$ defined by $F^v(1 - F) = z$, we can solve this for $\tau_J$:

$$\tau_J = \frac{(\alpha + 1)(\alpha + R_0)}{\alpha}F\left(\frac{2\alpha + 1}{(\alpha + 1)\left(\frac{\alpha}{R_0} + 1\right)}\left[\frac{\alpha^2}{(\alpha + 1)(\alpha + R_0)}\right]^\alpha, \alpha\right) - \alpha \tag{D8}$$

This allows us to compute $A_2$ and $A_3$

$$A_2 = \left.\frac{\ddot{J}}{J}\right|_{\tau = \tau_J} = \beta\dot{K} = \frac{\gamma^2(1 + \alpha)}{(\alpha + \tau_J)^2}\left(1 + \frac{\tau_J}{\alpha}\right)^{-2\alpha}\left[(R_0 - \tau)\left(1 + \frac{\tau_J}{\alpha}\right)^\alpha - R_0\right]$$
$$\left[-(1 + 2\alpha)R_0 + (\alpha + R_0)\left(1 + \frac{\tau_J}{\alpha}\right)^\alpha\right] \tag{D9}$$

$$A_3 = \left.\frac{\dddot{J}}{J}\right|_{\tau = \tau_J} = \beta\ddot{K} = \frac{\gamma^3(1 + \alpha)}{(\alpha + \tau_J)^3}\left(1 + \frac{\tau_J}{\alpha}\right)^{-3\alpha}\left[(R_0 - \tau_J)\left(1 + \frac{\tau_J}{\alpha}\right)^\alpha - R_0\right]$$
$$\left[-(\alpha + R_0)(\alpha + 2R_0 - \tau_J)\left(1 + \frac{\tau_J}{\alpha}\right)^{2\alpha}\right.$$
$$+ (1 + \alpha)R_0\left(1 + \frac{\tau_J}{\alpha}\right)^\alpha(3\alpha + 2(2 + \alpha)R_0 - (1 + 2\alpha)\tau_J)$$
$$\left. - 2(1 + \alpha)(1 + 2\alpha)R_0^2\right] \tag{D10}$$

The parameters $\lambda_0$, $\lambda_\infty$, $A_2$ and $A_3$ can be obtained from linear and cubic fits to the logarithm the number of daily reported cases $J_{\text{rep}}$. For these fits, time intervals corresponding to initial exponential growth ($T_i$), peak $T_p$ and final decay $T_f$ need to be defined. We use the time point $t_m^{\text{rep}}$, where $J_{\text{rep}}$ reaches its maximum as a reference point relative to which the intervals are given by:

$$T_i = [t_m^{\text{rep}} - 3\Delta t, t_m^{\text{rep}} - \Delta t] \tag{D11}$$

$$T_p = [t_m^{\text{rep}} - \Delta t, t_m^{\text{rep}} + \Delta t] \tag{D12}$$

$$T_f = [t_m^{\text{rep}} + \Delta t, t_m^{\text{rep}} + 3\Delta t] \tag{D13}$$

These time intervals are further reduced depending on the used data and $\Delta t$ such that all time points before the last day with $J_{\text{rep}} = 0$ prior to $t_m^{\text{rep}}$ and after the first day $J_{\text{rep}} = 0$ after $t_m^{\text{rep}}$

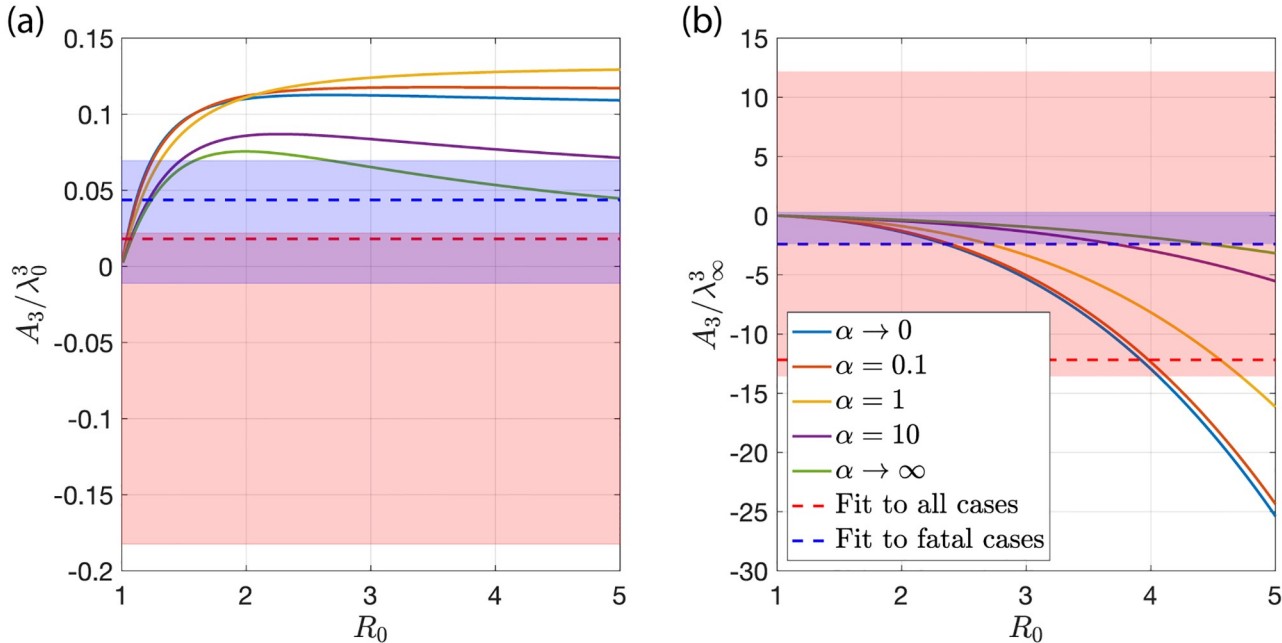

**Fig 7. Normalized coefficient $A_3 = \ddot{J}/J$ at the peak of new cases per day.** (a) The ratio $A_3/\lambda_0^3$ as a function of $R_0$ is shown for different values of $\alpha$. (b) The ratio $A_3/\lambda_\infty^3$ as a function of $R_0$ for the same values of $\alpha$. The dashed lines represent the values inferred from the data shown in Fig 3 for all cases (red) and fatal cases (blue). The shaded colored regions correspond to the uncertainties of the fits to the data.

are excluded. The fits in Fig 3 and the dashed horizontal lines in Figs 5 and 7 correspond to fits with $\Delta t = 19$ days. The shaded areas in Figs 5 and 7 depict the range of parameter values one obtains for fits with $10$ days $\leq \Delta t \leq 20$ days.

## Appendix E: Small $\alpha$ limit for heterogeneous populations

For small $\alpha$ the system reaches a well defined limiting dynamics that can be expressed analytically. We start from $I(\tau)$ for small $I_0/N$

$$\frac{I}{N} = 1 - \left(1 + \frac{\tau}{\alpha}\right)^{-\alpha} - \frac{\tau}{R_0} \tag{E1}$$

which for small $\alpha$ becomes

$$\frac{I}{N} \simeq \alpha \ln\left(1 + \frac{\tau}{\alpha}\right) - \frac{\tau}{R_0} \quad . \tag{E2}$$

The maximum of $I$ occurs at $\tau = \tau_I$ when $I' = 0$ or $\tau_I/\alpha \simeq R_0 - 1$. We thus have

$$\frac{I_{\max}}{N} \simeq \alpha\left(\ln R_0 + \frac{1}{R_0} - 1\right) \quad . \tag{E3}$$

Similarly, using $C_I/N = 1 - (1 + \tau_I/\alpha)^{-\alpha}$, we find for small $\alpha$

$$\frac{C_I}{N} \simeq \alpha \ln R_0 \tag{E4}$$

At long times, we have $I(\tau_\infty) = 0$, where $(1 + \tau_\infty/\alpha)^{-\alpha} = 1 - \tau_\infty/R_0$. For small $\alpha$ this implies $\alpha \ln(1 + \tau_\infty/\alpha) \simeq \tau_\infty/R_0$ and thus $S_\infty = (N - I_0)\bar{x}_\infty^\alpha$ and $\lambda_\infty = \beta \bar{x}_\infty - \gamma$, where

$$\bar{x}_\infty^{-1} = -R_0 W_{-1}\left(-\frac{e^{-1/R_0}}{R_0}\right) \quad . \tag{E5}$$

In the limit of small $\alpha$, $u = \tau/\alpha$ is finite. The limiting function $u(t)$ for small $\alpha$ can be expressed as

$$\int_0^u \frac{du'}{\ln(u'+1) - u'/R_0 + i_0} = \beta t \quad , \tag{E6}$$

where $i_0 = I/(\alpha N)$ in the limit $\alpha = 0$. The number of susceptible then becomes

$$\frac{S}{N} \simeq 1 - \alpha \ln(1 + u) \quad . \tag{E7}$$

Finally we discuss the maximum of the rate of new cases $J = J_{\max}$. We have $\dot{J}/J = \beta K$, where

$$K = \left(2 + \frac{1}{\alpha}\right)\left(1 + \frac{\tau}{\alpha}\right)^{-(\alpha+1)} - \frac{1}{R_0} + \frac{\alpha+1}{\alpha}\frac{1 - \tau/R_0}{1 + \tau/\alpha} \tag{E8}$$

At the maximum of $J$, $\tau = \tau_J$ with

$$(2\alpha + 1)\left(1 + \frac{\tau_J}{\alpha}\right)^{-\alpha} = \frac{\alpha}{R_0}\left(1 + \frac{\tau_J}{\alpha}\right) + (\alpha+1)\left(1 - \frac{\tau_J}{R_0}\right) \tag{E9}$$

Defining $\bar{x}_J = (1 + \tau_J/\alpha)^{-1}$ we have in the limit of small $\alpha$

$$\bar{x}_J = \exp\left(\frac{1}{R_0} - 1\right) \quad . \tag{E10}$$

The value of $J$ at the maximum is

$$\frac{J_{\max}}{N} = \alpha \gamma R_0 \bar{x}_J\left(-\ln \bar{x}_J - \frac{1 - \bar{x}_J}{R_0 \bar{x}_J}\right) \quad . \tag{E11}$$

We determine $A_2 = \ddot{J}/J = \beta^2 \dot{K}$ and $A_3 = \dddot{J}/J = \beta^3 \ddot{K}$, with $\dot{K}/\beta = K'I/N$ and $\ddot{K}/\beta^2 = K''I^2/N^2$. We then find

$$A_2 = -\gamma^2 R_0^2 \bar{x}_J^2\left(-\ln \bar{x}_J - \frac{1 - \bar{x}_J}{R_0 \bar{x}_J}\right) \tag{E12}$$

$$A_3 = 2\gamma^3 R_0^3 \bar{x}_J^3\left(-\ln \bar{x}_J - \frac{1 - \bar{x}_J}{R_0 \bar{x}_J}\right)^2 \quad . \tag{E13}$$

We also have

$$\frac{A_2^2}{A_3} = \frac{\gamma R_0 \bar{x}_J}{2} \tag{E14}$$

and

$$\frac{A_3^2}{A_2^3} = 4\left(-\ln \bar{x}_J - \frac{1 - \bar{x}_J}{R_0 \bar{x}_J}\right) \quad . \tag{E15}$$

## Appendix F: Mitigation in the heterogeneous SIR model

We now consider the case where the rate of infections $\beta(t)$ becomes time dependent because of overall changes of conditions such as seasonal effects or measures of social distancing. Using $I' = -S' + N/R_0$, we have $\lambda = \dot{I}/I = -\beta S'/N - \gamma$ and the reproduction number

$$R = -\frac{\beta}{\beta_0} R_0 \frac{S'}{N}. \tag{F1}$$

where $\beta_0 = \beta(t = 0)$ and $R_0 = \beta_0/\gamma$. The epidemic can be mitigated by a reduction of $\beta$ over time. However if the mitigation is relaxed the epidemic can grow again. As the epidemic advances, $\tau$ increases as $\dot{\tau} = \beta I/N$. Growth of infection number is no longer possible for $\tau > \tau_I$ with

$$-S'(\tau_I) = \frac{1}{R_0} \tag{F2}$$

Thus the condition $\tau > \tau_I$ defines herd immunity conditions where the epidemic can no longer grow. If mitigation sets in early, before $\tau = \tau_I$, the epidemic is slowed and it takes more time to reach herd immunity. in this case a new wave starts after mitigation is relaxed. If mitigation occurs for $\tau > \tau_I$, mitigation facilitates the decay of infections by reducing $\lambda_\infty = -(\beta_\infty/\beta_0)R_0 S'(\tau_\infty) - \gamma$ as compared to the value $\lambda_\infty = -R_0 S'(\tau_\infty) - \gamma$ without mitigation.

## Appendix G: Inferring $\beta(t)$ from reported cases

For a given time course of infections, there always exists a function $\beta(t)$ such that the SIR model follows this time course. We first consider the classic SIR model. A change in the rate of new infections $J = \beta IS/N$ can be decomposed in three different contributions,

$$\frac{\mathrm{d}}{\mathrm{d}t} \ln J = \frac{\dot{\beta}}{\beta} + \frac{\dot{I}}{I} + \frac{\dot{S}}{S}. \tag{G1}$$

In the case of an early mitigation, $S \approx N$ and thus $\dot{S}/S \approx 0$. Together with Eq (2), we find

$$\frac{\mathrm{d}}{\mathrm{d}t} \ln J = \frac{\mathrm{d}}{\mathrm{d}t} \ln \beta + \beta - \gamma. \tag{G2}$$

This provides a differential equation for $\ln \beta$ if $\ln J(t)$ is given, which does not require knowledge of the amplitude of $J$. We infer $\beta(t)$ for each day, using the initial value $\beta(0) = 0.48\text{days}^{-1}$ at March 15. We use an iterative scheme to calculate the rate for the next day as

$$\ln \beta(i+1) = \ln \beta(i) + \ln J_{\text{obs}}(i+1) - \ln J_{\text{obs}}(i) - \beta(i) + \gamma, \tag{G3}$$

where $\ln J_{\text{obs}}(i) = (1/7) \sum_{\Delta t = -3}^{3} J_{\text{rep}}(i + \Delta t)$ is a running average over seven days of the number of reported cases.

For the two scenarios of a later mitigation, the heterogeneous SIR model was considered with $J = \beta I(1 - I_0/N)(1 + \tau/\alpha)^{-(\alpha+1)}$. We then have

$$\frac{\mathrm{d}}{\mathrm{d}t} \ln J = \frac{\mathrm{d}}{\mathrm{d}t} \ln \beta + \frac{\mathrm{d}}{\mathrm{d}t} \ln I - \frac{\alpha+1}{\alpha}\left(1 + \frac{\tau}{\alpha}\right)^{-1} \frac{\beta I}{N} \quad . \tag{G4}$$

Again, this equation can be used to construct an iterative scheme to infer $\beta(t)$. For given initial number of infected individuals on March 15 $I(0)$, we can iteratively obtain the subsequent

values as

$$\ln \tau(i+1) \quad = \ln \tau(i) + \quad \frac{\beta(i)I(i)}{N\tau(i)}, \tag{G5}$$

$$\ln I(i+1) \quad = \ln I(i) + \quad \beta(i)(1 - \frac{I_0}{N})(1 + \frac{\tau(i)}{\alpha})^{-(\alpha+1)} - \gamma, \tag{G6}$$

$$\ln \beta(i+1) \quad = \ln \beta(i) + \quad \ln \frac{J_{\mathrm{obs}}(i+1)}{J_{\mathrm{obs}}(i)} - \ln \frac{I(i+1)}{I(i)}$$
$$+ \frac{\alpha+1}{\alpha}(1 + \frac{\tau(i)}{\alpha})^{-1} \frac{\beta(i)I(i)}{N} \quad . \tag{G7}$$

The starting value of $\tau(0)$, can be derived by inverting Eq (C7) for $I(\tau(0)) = I(0)$.

## Appendix H: Mobility data

Data concerning the changes in mobility of the population has been provided by Google [34]. The data reports the changes compared to a baseline of visits and length of stay at different places. The baseline depends on the specific day of the week and refers to the median value, for the corresponding day of the week, during the 5-week period Jan 3–Feb 6, 2020. Fig 8 shows these changes for Germany for a representative number of categories. These categories are defined in [34] as follows: "Grocery and pharmacy: Mobility trends for places like grocery markets, food warehouses, farmers markets, specialty food shops, drug stores, and pharmacies. Transit stations: Mobility trends for places like public transport hubs such as subway, bus, and train stations. Retail and recreation: Mobility trends for places like restaurants, cafes, shopping centers, theme parks, museums, libraries, and movie theaters. Residential: Mobility trends for

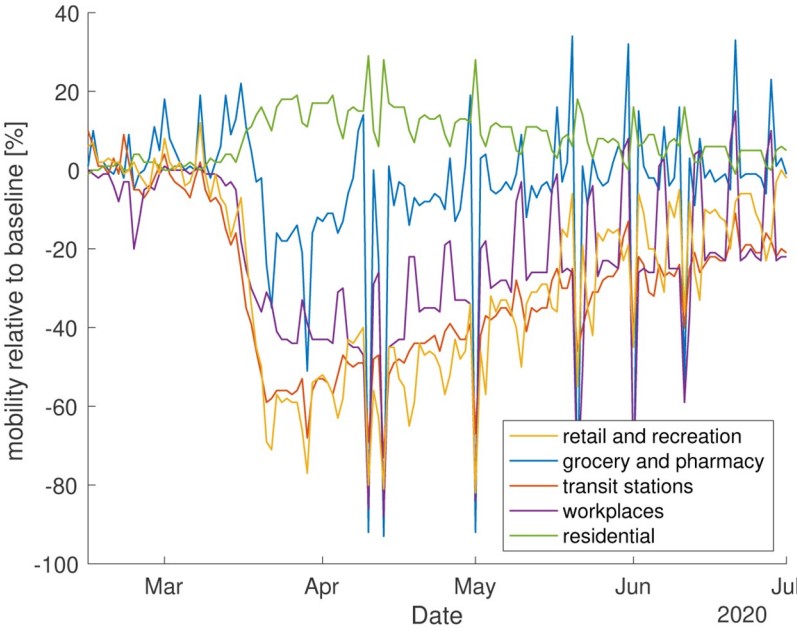

**Fig 8. Mobility changes for a representative set of commonly visited places in Germany up to July 1 2020 from [34].**

places of residence. Workplaces: Mobility trends for places of work. The residential category shows a change in duration while the other categories measure a change in total visitors."

## Author Contributions

**Conceptualization:** Jonas Neipel, Jonathan Bauermann, Stefano Bo, Tyler Harmon, Frank Jülicher.

**Funding acquisition:** Frank Jülicher.

**Investigation:** Jonas Neipel, Jonathan Bauermann, Stefano Bo, Tyler Harmon, Frank Jülicher.

**Project administration:** Frank Jülicher.

**Supervision:** Frank Jülicher.

**Writing – original draft:** Jonas Neipel, Jonathan Bauermann, Stefano Bo, Tyler Harmon, Frank Jülicher.

**Writing – review & editing:** Jonas Neipel, Jonathan Bauermann, Stefano Bo, Tyler Harmon, Frank Jülicher.

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
