## [Decision Letter · Decision Letter 0]

3 Sep 2020

PONE-D-20-24977

Power-Law Population Heterogeneity Governs Epidemic Waves

PLOS ONE

Dear Dr. Jülicher,

Thank you for submitting your manuscript to PLOS ONE. After careful consideration, we feel that it has merit but does not fully meet PLOS ONE’s publication criteria as it currently stands. Therefore, we invite you to submit a revised version of the manuscript that addresses the points raised during the review process.

Comments of reviewers are valuable and give you an opportunity to improve the paper.

We look forward to receiving your revised manuscript.

Kind regards,

Vygintas Gontis, Ph.D.

Academic Editor

PLOS ONE

Journal Requirements:

2. Thank you for inlcuding your funding statement;

Reviewers' comments:

Reviewer's Responses to Questions

**Comments to the Author**

1. Is the manuscript technically sound, and do the data support the conclusions?

Reviewer #1: Yes

Reviewer #2: Yes

2. Has the statistical analysis been performed appropriately and rigorously? 

Reviewer #1: Yes

Reviewer #2: Yes

3. Have the authors made all data underlying the findings in their manuscript fully available?

Reviewer #1: Yes

Reviewer #2: Yes

4. Is the manuscript presented in an intelligible fashion and written in standard English?

Reviewer #1: Yes

Reviewer #2: Yes

5. Review Comments to the Author

Reviewer #1: In the paper under review, the authors give a detailed study of the classical SIR model for epidemics in the case when the heterogeneity of the susceptible population is allowed. The main focus is made on the case where the initial susceptibility distribution has gamma density with parameter $\\alpha$. This, in particular, leads to the time-dependent reproduction number $R(t)$, decreasing as hyperbolic function, i.e.\\ $R(t)=(\\bar x(t))^{1+\\alpha} R_0$. The authors provide quantitative properties of the model, such as the herd immunity level, the final size of epidemics, etc. An important conclusion is that, in the heterogeneous population, the herd immunity level can be much lower than in homogeneous case (typically 60\\%).

In my opinion, the subject and results of the paper are very interesting, the paper is well written and I recommend it for publishing in the journal after minor revision. Attached, please find a list of comments.

Reviewer #2: In this manuscript, the authors propose a generalized SIR model taking into account the heterogeneity of the population, i.e., a distribution of the susceptibility of being infected, in terms of a parameter alpha, which is the power-law exponent of the susceptibility distribution when small values of alpha are considered. In other words, when alpha -> infinity, the classical SIR model is recovered and for the case of alpha -> 0, the heterogeneity of the population is incorporated into the SIR model. The key-result of their work is that the population herd immunity is earlier achieved in the case of a heterogeneous population, implying in a lower number of infected people and fatalities. The authors employ their model to analyze the case of the Covid-19 spread in Germany and discuss the importance of taking the population's heterogeneity in the effectiveness of mitigation actions, such as social distancing. Below, I raise some points to be addressed:

- In panel (a) and (d) of Fig. 1, it is not quite clear to me why the number of infected people is lower for the heterogeneous SIR model. Also, what is the justification for using the specifically values of R0 = 2.5, gamma = 0.13 day^-1 and alpha = 0.1? I suggest that the authors make these points clear;

- For future works, I believe it would be interesting to explore the very same analysis here employed in terms of the heterogeneity of the population in the light of the SIRS model, since it would be interesting to analyse how the population's susceptibility distribution is affected in the case where Recovered people can become susceptible of being infected again. Perhaps it would interesting to mention this in the main text;

- On page 6, the authors mention about the Lambert W function and points to a more detailed discussion about it in Appendix A. At this point, I believe it is worth adding a couple of references for clarity both in the main text and in Appendix A about the Lambert W function, just for the sake of completeness;

- The authors should correct on page 3, third paragraph, first sentence, the typo "the" twice in the sentence "In the heterogeneous SIR model proposed here, the qualitative behaviors of the the epidemic wave are unchanged.";

- I suggest that the first sentence of page 8 "Eq. (9) can be then be written as" to be corrected to "Eq. (9) can then be written as...";

- On page 8, the authors write "The dynamics of epidemic waves depends on the shape of the initial distribution s0(x). Here, we consider distributions that have the special property of shape invariance under the dynamics of epidemics. This property is satisfied by a gamma distribution". I suggest that the authors state if only the gamma distribution satisfies such a condition and, if it is not the case, then it would be interesting to add a sentence about the consideration of other distributions as well;

- In Fig 4 panel (a), I suggest that the authors state why their solutions of the heterogeneous SIR model do not incorporate the initial increase in the number of infected people and the small increase between Jun and Jul for both the number of infections and fatalities. The same holds for panel (c) in the case of the initial growth of both the cumulative number of infected people and fatalities. In panel (f), it would be interesting to discuss what is the meaning of tau saturating at the specific value of ~0.2, as well as about the meaning of the average susceptibility x saturating close to the \\tau curve? What does this mean? I suggest that the authors briefly discuss about these points;

- Based on their discussions, the achievement of the herd immunity is key regarding the fade of the disease spread. Based on their discussion about and heterogeneous population, do the authors have any suggestions for public policies in order to minimize the number of infections and, consequently, the number of fatalities?;

- On page 18, the authors write "We show that as a result of strong population heterogeneity (small alpha), the wave peaks when only a small minority of individuals have been infected, see Fig. 1 (d)-(f)." This is indeed true. However, upon analysing panels (a), (d), and (g) of Fig. 6, one notices that the model solution fit of the data (red solid line) present lower reported cases decrease rate than in the scenario without mitigation (red dotted line). This is particularly true after June. How can this be explained? It seems that, although the maximum is earlier achieved, the decrease rate is lower than in the case without mitigation. I suggest that the authors include a few sentences to discuss about this;

- Section "Discussion" seems more like "Conclusions and Perspectives";

- I believe that the number of references could be improved in this work since there are a lot of discussions throughout the manuscript that deserves more important references.

In summary, the work is relevant since in reality not everyone is equally susceptible to being infected and thus the authors' consideration of a susceptibility distribution among the population is solid and can indeed improve the understanding of the epidemics dissemination. I do recommend publication after minor revisions.

6. PLOS authors have the option to publish the peer review history of their article (what does this mean?). If published, this will include your full peer review and any attached files.

Reviewer #1: No

Reviewer #2: No

---

## [Author Response · Author response to Decision Letter 0]

9 Sep 2020

Response to Referee 1

>In the paper under review, the authors give a detailed study of the classical SIR 

>model for epidemics in the case when the heterogeneity of the susceptible 

>population is allowed. The main focus is made on the case where the initial 

>susceptibility distribution has gamma density with parameter $\\alpha$. 

>This, in particular, leads to the time- dependent reproduction number $R(t)$, 

>decreasing as hyperbolic function, i.e.\\ $R(t)=(\\bar x(t))^{1+\\alpha} R_0$. 

>The authors provide quantitative properties of the model, such as the herd 

>immunity level, the final size of epidemics, etc. An important conclusion is that, 

>in the heterogeneous population, the herd immunity level can be much lower 

>than in homogeneous case (typically 60\\%). 

>

>In my opinion, the subject and results of the paper are very interesting, the 

>paper is well written and I recommend it for publishing in the journal after 

>minor revision. Attached, please find a list of comments. 

We thank the referee for a careful reading of our manuscript

>1. In the models for SARS-CoV-2 epidemics, the SEIR model is in widespread 

>use, where E stands for the ”exposed” compartment. I wonder, if it is possible 

>to relate it with the generalized SIR model in (1)–(2) with time- varying 

>parameters?  

In our work we focus on the simpler SIR model where the distinction between 

exposed and infected is not made. The additional “exposed” state E in the SEIR 

model introduces a time delay between exposure to a pathogen and the 

onset of infectiousness. This extra delay due to the exposed state could be 

captured by an exponential memory kernel in Eq (2) with an extra relaxation time 

describing the delay. However, the main properties of the epidemic wave such as 

herd immunity levels are only weakly affected by the extra short delay in the SEIR 

model. 

In our work we focus on the SIR model because it captures essential features of 

an epidemics in a minimal model. In our revised manuscript we now relate to the 

SEIR model in the discussion.

>2. The model in eqs. (1)–(2) is not very clearly explained. The classical SIR 

>model assumes three components – susceptible (S), infected (I) and recov- 

>ered (R). Whereas eqs. (1)–(2) of the paper does not include part R. Also, 

>it is slightly unclear the meaning of x ¯. It would be useful to explain what does 

>it mean ”dimensionless average susceptibility”. Usually the coefficient at IS/N 

>is called ”infection rate” and can admit values larger (or smaller) than 1. 

 

We use the symbol R for the time dependent reproduction number rather than 

the number of recovered individuals. Note that the number of recovered 

individuals is given by #recovered = N-S-I. We now add this information. In the 

revised manuscript we have rewritten the explanation of the model in order to 

be more clear about beta and \\bar x and we now call beta the infection rate.

>3. p. 5. It is mentioned that time-varying β could correspond to seasonal 

>changes or mitigation measure. For illustrative purposes, it would be nice to 

>see concrete forms of β(t) in such cases.  

While in the first parts of the manuscript we consider constant beta, we 

discuss in section III.C different mitigation scenarios where we choose specific 

forms of beta(t) in order to discuss effects of mitigation during this year’s 

SARS-Cov2 epidemic, see Fig. 6.

>4. It is well-known that SIR-type compartment models are rather sensitive to 

>initial conditions. It would would be good to see some discussion on this 

>sensitivity. At least, how the graphs will change if you take other values than 

>I0 = 10.  

The initial condition I0 is not important for our work and a change in I0 would 

essentially only shift the absolute time of the initial point if the time of the 

wave peak is known. Once the wave is progressing and reaches its peak, its 

behaviour depends only very weakly on the initial conditions if the initial number 

of infected is small compared to the population size. Note that in the appendices 

we do provide exact expressions for many quantities as a function of I0/N, 

revealing the role of initial conditions. However for all plots and results shown 

in the manuscript the initial number I0 is completely unimportant. Because 

the plots for other choices of I0 would not be distinguishable from the ones 

shown we prefer not to show plots with other values of I0.

Another reason not to discuss different I0 is that at the early time of the 

epidemics when I(t) went from say 10 to 20 we have absolutely no information 

about the epidemic wave and therefore it does not to help discussing wether 

it started with I0=2 very early or with I0=15 several days later.

>5. p. 7, l. -9. Since s depends on both x and t, I would recommend to write 

>S(t) = ∫ \\int_0^∞ dxs(x, t). Similarly, in eq. (10), as the right-hand side 

>depends on t, better to write x ¯(t) = (1/S(t))\\int_0^∞ dx xs(x, t). 

We have changed these expression as the referee suggests.

>6. p. 7, l. -2. The meaning of the variable τ (”a measure for how far the 

>epidemic has advanced”) should be explain better. 

The variable tau emerges from a trick to solve the equations. It does not have 

a physical meaning but it has some clear and important properties. 

In the revised manuscript we now explain better the variable tau.

>7. p, 8, eq. (12). Strictly speaking, eq. (12) gives a density function, not 

>distibution. Also, you may wish to add that α > 0 and to write a precise form 

>s_0(x)=…(The symbol ’∼’ usually means asymptotic equivalence.) Also, I 

>wonder if it would be possible to introduce additional flexibility in the initial 

>susceptibility by introducing two-parameter gamma distribution with 

>density s_0(x)=…..

We agree with the referee and have added that alpha >0. Note that we 

thought about all these points carefully and we have good reasons to present 

the work in the way we did. We have used the term distribution in the context 

of our aim to make the paper more easily accessible for a broader readership 

including non-experts for whom the term probability distribution is usually 

better known. For this reason we have also kept the equations in the main 

part of the manuscript rather light, but we provide all the precise and full 

expressions in the Appendices for expert readers. The appendices carry a 

lot of substance and should not be seen as secondary.

Note that the precise form of the initial density function s_0 is provided in 

Appendix C in Eq. (C1). In our revised manuscript we now refer to the 

appendix to clarify what the normalization factor is.

We have checked that the full 2-parameter gamma distribution does not 

provide any further flexibility. The reason is that we choose without of loss 

of generality \\bar x=1 at the initial time point. With the second parameter 

of the gamma distribution we can change this initial value but this change 

can be absorbed by changing the infection rate beta. So in fact one can 

see beta as the second parameter. But since beta already exists in the 

classical SIR model we keep it and only add one new parameter alpha 

describing the ratio of variance and mean.

>Minor comments: 

>p. 3, l. -11. Delete ’the’. 

 

Done

>p. 8, eq. (13). Please add more details how this equality is obtained. 

We added some information and now refer to Appendix C for details.  

p. 9. Please add more details how eqs. (18) and (19) are obtained.  

We added a reference to the Appendix C where the details are provided.

>p. 22. It seems that the Lambert W -function should be defined by 

>W (z)eW (z) = z.  

Done

Response to Referee 2

>In this manuscript, the authors propose a generalized SIR model taking 

>into account the heterogeneity of the population, i.e., a distribution of the 

>susceptibility of being infected, in terms of a parameter alpha, which is 

>the power-law exponent of the susceptibility distribution when small values 

>of alpha are considered. In other words, when alpha -> infinity, the 

>classical SIR model is recovered and for the case of alpha -> 0, the 

>heterogeneity of the population is incorporated into the SIR model. The 

>key-result of their work is that the population herd immunity is earlier 

>achieved in the case of a heterogeneous population, implying in a lower 

>number of infected people and fatalities. The authors employ their model 

>to analyze the case of the Covid-19 spread in Germany and discuss the 

>importance of taking the population's heterogeneity in the effectiveness 

>of mitigation actions, such as social distancing. 

>Below, I raise some points to be addressed:

>- In panel (a) and (d) of Fig. 1, it is not quite clear to me why the number 

>of infected people is lower for the heterogeneous SIR model. Also, what 

>is the justification for using the specifically values of R0 = 2.5, 

>gamma = 0.13 day^-1 and alpha = 0.1? I suggest that the authors 

>make these points clear;

The figure just shows the fact that the number of infected people is lower 

in the heterogeneous SIR model to clearly show this point. The reasons are 

the lowered herd immunity levels given by Eq (19) resulting from the drop 

in \\bar x as shown in Fig. 1 f. In appendix C we provide an exact analysis 

of the nonlinear dynamics that reveals these surprising properties. 

In order to explain this better, we now clarify after Eq (22) that the drop 

of \\bar x is the reason for a reduced herd immunity level and resulting 

lower infection numbers.

The parameter values are here just for illustrative purposes. The qualitative 

behaviors do not depend on the parameter choice within broad ranges. 

However the values chosen are rounded versions of values we found are 

relevant to the current SARS-Cos2 epidemics are typical values used in the 

current epidemics. gamma = 0.13 corresponds to individuals being 

infectious during one week, and this parameter does not affect the shape 

but only the duration of the wave. alpha=0.1 was chosen so that the 

difference between panels (a) and (d) is clearly visible but such that the 

maximum of I(t) can still be seen in (d). 

>- For future works, I believe it would be interesting to explore the very 

>same analysis here employed in terms of the heterogeneity of the population 

>in the light of the SIRS model, since it would be interesting to analyse how 

>the population's susceptibility distribution is affected in the case where 

>Recovered people can become susceptible of being infected again. 

>Perhaps it would interesting to mention this in the main text;

There are many open and interesting questions that one can address 

with our approach. We agree that it will be very interesting to look into 

effects where recovered individuals becomes infected again. In the 

revised manuscript we now mention the SIRS model in the discussion.

>- On page 6, the authors mention about the Lambert W function and 

>points to a more detailed discussion about it in Appendix A. At this point, 

>I believe it is worth adding a couple of references for clarity both in the 

>main text and in Appendix A about the Lambert W function, just for the 

>sake of completeness;

We have added a reference on the Lambert W function.

>- The authors should correct on page 3, third paragraph, first sentence, 

>the typo "the" twice in the sentence "In the heterogeneous SIR model 

>proposed here, the qualitative behaviours of the the epidemic wave are 

>unchanged.";

Thanks - corrected

>- I suggest that the first sentence of page 8 "Eq. (9) can be then be 

>written as" to be corrected to "Eq. (9) can then be written as…";

 Thanks - done

>- On page 8, the authors write "The dynamics of epidemic waves depends 

>on the shape of the initial distribution s0(x). Here, we consider distributions 

>that have the special property of shape invariance under the dynamics of 

>epidemics. This property is satisfied by a gamma distribution". I suggest that 

>the authors state if only the gamma distribution satisfies such a condition and, 

>if it is not the case, then it would be interesting to add a sentence about the 

>consideration of other distributions as well;

To our knowledge the gamma distribution is the only distribution that is shape 

invariant under the dynamics. However to be cautious we avoid claiming this 

fact as we do not have a proof. Our work and our results do not depend on 

the gamma distribution being the only shape invariant distribution.

>- In Fig 4 panel (a), I suggest that the authors state why their solutions of 

>the heterogeneous SIR model do not incorporate the initial increase in the 

>number of infected people and the small increase between Jun and Jul for 

>both the number of infections and fatalities. The same holds for panel (c) in 

>the case of the initial growth of both the cumulative number of infected 

>people and fatalities.

In Fig. 4 we simply compare fits of our model to data. The fits show that 

the data based on reported cases that lead to deaths (blue), which probably 

measures more reliably the progression of serious illnesses, is better 

captured by the model than simply using the reported number of reported 

cases (red). A possibility for the difference to the data early and in June is 

that the blue data gives a better representation of the epidemics than the 

red data. However we have refrained form making such statements because 

all available data have problems and there are many reasons why there 

could be serious biases and errors in the data. 

In our manuscript we note on p. 13 that it is surprising that the model fits 

both types of data rather well even though we only have three time 

independent parameters (note the classical SIR model cannot account for 

this data). 

>In panel (f), it would be interesting to discuss what is the meaning of tau 

>saturating at the specific value of ~0.2, as well as about the meaning of 

>the average susceptibility x saturating close to the \\tau curve? What does 

>this mean? I suggest that the authors briefly discuss about these points;

The final value of tau is explained in Appendix C where we show that tau 

reaches a fixed final value when the epidemics dies out that is described 

exactly by the implicit equation (C11) in the revised manuscript. In our 

revised manuscript we now mention in the main text on p. 7 that tau 

reaches a final value and we add a reference to appendix C when discussing 

panel (f) of Fig. 4. Note that the relationship between tau and \\bar x is 

given in Eq. (14). The similarity of \\bar x and \\tau at the end of the wave 

is a coincidence and consistent with Eq. (14).

>- Based on their discussions, the achievement of the herd immunity is key 

>regarding the fade of the disease spread. Based on their discussion about 

>and heterogeneous population, do the authors have any suggestions for 

>public policies in order to minimize the number of infections and, consequently, 

>the number of fatalities?;

We have on purpose refrained from a discussion of public policy. Here we 

want to focus on the concepts and the science. The science presented here 

is clear and rigorous. Implications for policy are much less rigorous and depend 

on many other factors and can be coloured by opinions. We think that our 

work has many implications for public policy and that it will stimulate and be 

useful for future discussions but we do not want to weaken our work by 

adding elements that are uncertain.

>- On page 18, the authors write "We show that as a result of strong 

>population heterogeneity (small alpha), the wave peaks when only a small 

>minority of individuals have been infected, see Fig. 1 (d)-(f)." This is indeed 

>true. However, upon analysing panels (a), (d), and (g) of Fig. 6, one notices 

>that the model solution fit of the data (red solid line) present lower reported 

>cases decrease rate than in the scenario without mitigation (red dotted line). 

>This is particularly true after June. How can this be explained? It seems that, 

>although the maximum is earlier achieved, the decrease rate is lower than 

>in the case without mitigation. I suggest that the authors include a few 

>sentences to discuss about this;

It is correct that in the case of the epidemics that is dying out because of 

mitigation (Fig. 6), we find that the rate of decay of cases is slower than without 

mitigation. This is similar to the idea to “flatten the curve”, i.e. mitigation 

reduces the maximal number of infections but broadens the wave and thus 

makes it slower. However this feature is not completely general and we prefer 

not to enter this discussion.

>- Section "Discussion" seems more like "Conclusions and Perspectives";

We have changed the discussion to “Conclusions and Perspectives”

>- I believe that the number of references could be improved in this work 

>since there are a lot of discussions throughout the manuscript that deserves 

>more important references.

We have now added reference [28] on Lamberts W function and Ref. [35-37] 

for the SEIR and the SIRS model. We think that all statements that need backing 

by references have been referenced. 

>In summary, the work is relevant since in reality not everyone is equally 

>susceptible to being infected and thus the authors' consideration of a susceptibility 

>distribution among the population is solid and can indeed improve the 

>understanding of the epidemics dissemination. I do recommend publication after 

>minor revisions.

We thank the referee for a careful reading of our manuscript

---

## [Editor Report · Decision Letter 1]

11 Sep 2020

Power-Law Population Heterogeneity Governs Epidemic Waves

PONE-D-20-24977R1

Dear Dr. Jülicher,

We’re pleased to inform you that your manuscript has been judged scientifically suitable for publication and will be formally accepted for publication once it meets all outstanding technical requirements.

Kind regards,

Vygintas Gontis, Ph.D.

Academic Editor

PLOS ONE

Additional Editor Comments (optional):

Congratulations, After careful analysis of the article revised version and Response to Reviewers, we have decided to accept the paper for the publication in Plos One.
---

## [Editor Report · Acceptance letter]

17 Sep 2020

PONE-D-20-24977R1 

Power-Law Population Heterogeneity Governs Epidemic Waves 

Dear Dr. Jülicher:

I'm pleased to inform you that your manuscript has been deemed suitable for publication in PLOS ONE. Congratulations! Your manuscript is now with our production department. 

Kind regards, 

on behalf of

Dr. Vygintas Gontis 

Academic Editor

PLOS ONE